# Natural killer (NK) cell-derived extracellular-vesicle shuttled microRNAs control T cell responses

**Sara G Dosil**[1,2,3], **Sheila Lopez-Cobo**[4†], **Ana Rodriguez-Galan**[1,2,3], **Irene Fernandez-Delgado**[1,2,3], **Marta Ramirez-Huesca**[1,2,3], **Paula Milan-Rois**[5], **Milagros Castellanos**[5], **Alvaro Somoza**[5], **Manuel José Gómez**[2], **Hugh T Reyburn**[4], **Mar Vales-Gomez**[4], **Francisco Sánchez Madrid**[1,2,3,6]*, **Lola Fernandez-Messina**[1,2,3,6]*

[1]Immunology Service, Hospital de la Princesa, Universidad Autónoma de Madrid, Instituto Investigación Sanitaria Princesa, Madrid, Spain; [2]Intercellular Communication in the Inflammatory Response. Vascular Pathophysiology Area, National Center for Cardiovascular Research (CNIC), Madrid, Spain; [3]Universidad Autónoma de Madrid, Madrid, Spain; [4]Department of Immunology and Oncology, National Centre for Biotechnology, Spanish National Research Council, Madrid, Spain; [5]Instituto Madrileño de Estudios Avanzados en Nanociencia (IMDEA Nanociencia) & Nanobiotecnología (IMDEA-Nanociencia), Unidad Asociada al Centro Nacional de Biotecnología (CSIC), Madrid, Spain; [6]CIBER de Enfermedades Cardiovasculares (CIBERCV), Madrid, Spain

*For correspondence:
fsmadrid@salud.madrid.org
(FSM);
lfernandezmessina@gmail.com
(LF-M)

Present address: †INSERM U932, Institut Curie, PSL Research University, Paris, France

Competing interest: The authors declare that no competing interests exist.

**Abstract** Natural killer (NK) cells recognize and kill target cells undergoing different types of stress. NK cells are also capable of modulating immune responses. In particular, they regulate T cell functions. Small RNA next-generation sequencing of resting and activated human NK cells and their secreted extracellular vesicles (EVs) led to the identification of a specific repertoire of NK-EV-associated microRNAs and their post-transcriptional modifications signature. Several microRNAs of NK-EVs, namely miR-10b-5p, miR-92a-3p, and miR-155-5p, specifically target molecules involved in Th1 responses. NK-EVs promote the downregulation of *GATA3* mRNA in CD4+ T cells and subsequent *TBX21* de-repression that leads to Th1 polarization and IFN-γ and IL-2 production. NK-EVs also have an effect on monocyte and moDCs (monocyte-derived dendritic cells) function, driving their activation and increased presentation and costimulatory functions. Nanoparticle-delivered NK-EV microRNAs partially recapitulate NK-EV effects in mice. Our results provide new insights on the immunomodulatory roles of NK-EVs that may help to improve their use as immunotherapeutic tools.

## Editor's evaluation

This report identified NK-extracellular-vesicle (NK-EV)-associated microRNAs and characterized them by small RNA next-generation sequencing. They found that NK-EVs promote Th1 polarization and activation of monocyte and moDCs. The findings are potentially important for understanding NK cell function.

## Introduction

Extracellular vesicles (EVs) are key mediators of cell-to-cell communication and play a crucial role in the regulation of immune responses (*Fernández-Messina et al., 2015*). EVs, for example exosomes, microvesicles, and apoptotic bodies, carry not only bioactive molecules such as proteins, carbohydrates, lipids, but also genetic information, including microRNAs (miRNAs) (*Colombo et al., 2014*). These miRNAs can be transferred among cells and modulate gene expression in the recipient cell.

Natural killer (NK) cells recognize and kill cells undergoing different types of stress, including aging, malignant transformation, and pathogenic infection. NK cells also modulate immune responses, particularly T cell function. NK cells promote T cell differentiation, proliferation, and cytokine production (*Crouse et al., 2015*). These effects are mediated by direct interactions between NK and T lymphocytes; but also indirectly through their effect on antigen-presenting cells and the secretion of soluble factors.

Every immune cell releases EVs, including NK cells. Recent evidence shows that NK-derived EVs (NK-EVs) can exert antitumoural functions. NK cell-derived exosomes mediate cytolytic effects on melanoma (*Zhu et al., 2017*) and neuroblastoma cells (*Shoae-Hassani et al., 2017*). In addition, NK cell-derived exosomes contain cytotoxic molecules (*Wu et al., 2019*), such as Fas-L and perforin, which induce tumour cell apoptosis (*Lugini et al., 2012*); or DNAM1 (*Di Pace et al., 2020*), which is also involved in exosome-mediated antitumoural responses, as revealed by experiments using blocking antibodies. Proteomic analyses of NK-EVs have identified additional effector candidates that may induce tumour cell death, including TRAIL, NKG2D, or fibrinogen (*Choi et al., 2020*) . Thus, increasing evidence supports the potential use of cytotoxic cell-derived EVs, including NK cells and also cytotoxic T lymphocytes (CTLs) as therapeutic agents (*Del Vecchio et al., 2021*). Although most studies to date have highlighted the potential role of NK-EVs in cancer, they can be potentially used to modulate other biological processes and pathologies. In particular, NK-EVs have a beneficial effect in lung injury recovery after *Pseudomonas aeruginosa* infection (*Jia et al., 2020*). NK-EVs curb CCL4-induced liver fibrosis in mice by inhibiting TGF-β1-induced hepatic stellate cells activation (*Wang et al., 2020*). Also, miR-207-containing NK-EVs alleviate depression-like symptoms in mice (*Li et al., 2020*).

Interestingly, recent studies have identified the role of biologically active NK-EV miRNAs, such as miR-186, which impaired neuroblastoma tumour growth and inhibited immune escape mechanisms by targeting the TGF-β pathway (*Neviani et al., 2019*); or miR-3607-3p, which inhibited pancreatic cancer, presumably by targeting its putative target IL-26 (*Sun et al., 2019*). These reports suggest the modulatory role of NK-EV-miRNAs to inhibit cancer progression. However, little is known regarding the overall small RNA composition of human NK-EVs, and their impact on the regulation of immune responses remains far from being fully elucidated.

In this study, we have set up a model to study human primary NK-EVs. We have analysed the small RNA content of resting and in vitro-activated NK cells and their secreted EVs by next-generation sequencing (NGS). We show that NK-EVs have specific miRNA repertoire and post-transcriptional modification (PtM) patterns, which differ from that of their parental cells. Analyses of the NK-EV miRNA signature revealed an enrichment of Th1 function-related miRNAs that target key T cell mRNA molecules. One example is *GATA3*, as its downregulation leads to T-bet de-repression. Other identified miRNAs are involved in dendritic cell (DC) presentation and costimulatory functions. We further show that NK-EVs promote T cell activation followed by IFN-γ and IL-2 production and induce DC expression of MHC-II and CD86 in DCs. Finally, in vivo nanoparticle-based delivery of the NK-EV-enriched miRNAs miR-10b-5p, miR-92a-3p, and miR-155-5p partially reproduces the effects of NK-EV treatment in T cell responses, suggesting an involvement of these miRNAs in effector functions. These findings shed light on the antitumour effects of NK-EVs, that among other effects, skew the balance towards inflammatory Th1 T cells, reinvigorating antitumoural responses and, therefore, support their use as immunotherapeutic tools.

## Results

### NK-EVs bear a specific miRNA signature different from their secreting cells

To identify the miRNA repertoire of resting- and in vitro-activated NK cells and their load into EVs, we set up the protocol shown in *Figure 1—figure supplement 1A*. Briefly, NK cells were enriched from

isolated peripheral blood mononuclear cells (PBMCs) of human healthy donor's buffy coats by adding a mixture of IL-12 and IL-18 together with irradiated feeder cells. Six days after, cells were kept in culture with IL-2 for a further 48 hr, in the absence of feeder cells, and NK cell expansion was tested by flow cytometry before and after NK cell isolation (*Figure 1—figure supplement 1B*). EVs accumulated for 72 hr from $15 \times 10^6$ NK cells were purified by serial centrifugation as previously described (*Ashiru et al., 2010*) and tracked (EV-TRACK ID: EV210234). RNA was isolated from resting NK cells (directly isolated without being cultured), in vitro-activated NK cells, and small EVs released from activated NK cells (*Figure 1—figure supplement 1A,C*). Vesicles isolated from activated NK cells were characterized using NanoSight showing a mode size ranging from 165 to 209 nm among donors (*Figure 1—figure supplement 1D-F*). Also, biochemical analyses showed the NK-EV expression of the exosomal markers CD63 and Tsg101 and the absence of the EV-excluded marker Calnexin (*Figure 1— figure supplement 1G*). In line with this, electron microscopy of isolated NK-EVs confirmed the presence of nano-sized membranous vesicles with their typical cup-shaped morphology (*Figure 1—figure supplement 1H*). Analyses of the small EVs released from either resting or activated NK cells showed increased EV release upon activation (*Figure 1—figure supplement 1I,J*). Indeed, EVs released from resting NK cells were barely detected using nanoparticle-tracking analysis (NTA) and dot blot techniques, for this reason they were not included in the study.

Small RNA sequencing analyses showed that resting and activated cells display a distinct miRNA profile that differs to that of cultured activated NK cells-secreted EVs, as shown in heatmap and principal component plots (*Figure 1*, *Figure 1—figure supplement 1K*, and *Supplementary file 1Supplementary file 1*). A total of 130 miRNAs were differentially expressed between activated NK cells and NK-EV released miRNAs; 96 miRNAs being enriched in NK-EVs and 34 miRNAs being more abundant in NK cells than in NK-EVs (*Figure 1B*). Such differential expression suggested the existence of mechanisms of specific miRNAs sorting into EVs derived from NK cells, in agreement with published data (*Mateescu et al., 2017*; *Villarroya-Beltri et al., 2013*; *Szostak et al., 2014*; *Batagov et al., 2011*; *Mittelbrunn et al., 2011*). Also, the repertoire found in resting NK cells was different from that of in vitro-activated NK cells (*Figure 1A*, *Figure 1—figure supplement 2*, and *Supplementary file 1*). A total of 70 miRNAs were upregulated upon activation, while 98 miRNAs decreased their levels after cytokine stimulation (*Figure 1—figure supplement 2A*). These data are in agreement with extensive miRNA remodelling occurring upon T lymphocyte activation (*Rodríguez-Galán et al., 2021*).

Unbiased analysis of the pathways potentially affected by miRNAs shuttled into NK-EVs using the ingenuity pathway analysis (IPA) revealed that the putative mRNA targets of NK-EV-enriched miRNAs were involved in cellular development and movement, cell growth and proliferation, cell death, survival, and cell cycle (*Figure 1C*). In silico mRNA target analyses for NK-EV-miRNAs identified putative target molecules related to immune signalling and Th1 responses, targeting CD4[+] T lymphocytes and DCs among other immune effectors (*Supplementary file 2*). Selected miRNAs identified in this screening (*Figure 1D*, and *Figure 1—figure supplement 2B, C*) were validated by qPCR (*Figure 1E and F*).

## PtMs in NK cells and EV miRNAs

Since changes in the miRNA repertoire during lymphocyte activation have been linked to their PtMs (*Rodríguez-Galán et al., 2021*; *Gutiérrez-Vázquez et al., 2017*), we analysed the global PtMs signatures of miRNAs from resting NK cells, activated NK cells, and released NK-EVs (*Figure 2A and B* and *Figure 2—figure supplement 1*). These analyses revealed a complex pattern of miRNA PtMs in human NK cells and released small EVs. Most PtMs appeared at the edges of the canonical sequence, particularly at the terminal nucleotide (position 0) and the two flanking nucleotide positions (–1 and +1). The 3p-end of miRNAs was significantly more prone to PtMs (*Figure 2A*) than the 5p-end in every sample (*Figure 2—figure supplement 1A-C*). Also, non-templated mono-additions of nucleotides were much more abundant than poly-additions (*Figure 2C*, and *Figure 2—figure supplement 1C*). Thus, we focused our analyses on the 3p-end miRNA's PtMs, in particular single nucleotide additions accumulated in positions –1 to +1. Interestingly, miRNAs from the different fractions (resting NK cells, activated NK cells, and NK-EVs) exhibited clear PtM signature differences (*Figure 2B and C*). Cytosine addition was the most common modification, followed by adenine terminal insertions. Upon NK cell activation, miRNAs displayed reduced adenylation (positions –1, +1) and cytosylation (positions –1, +1). Comparing activated NK cells

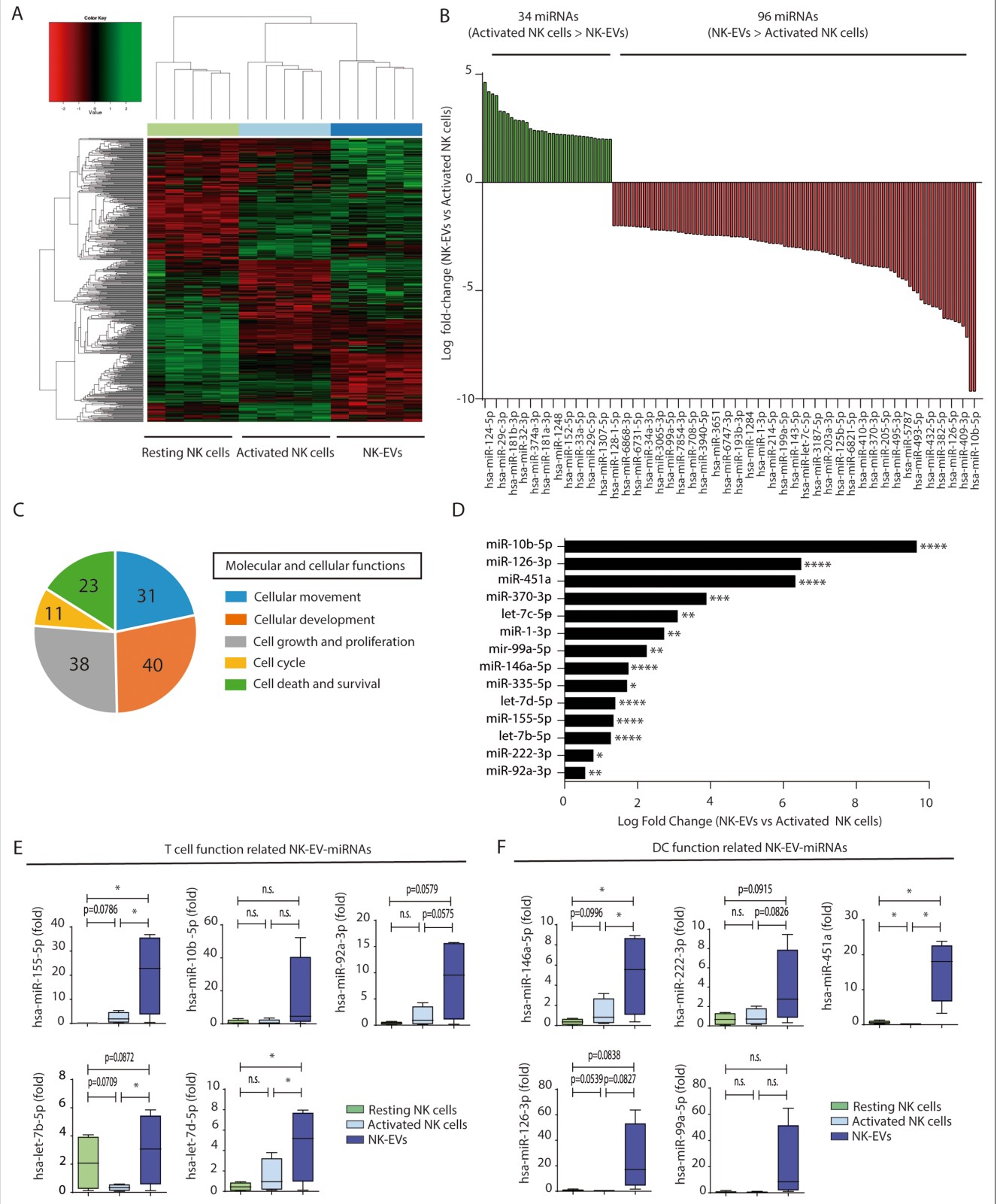

**Figure 1.** Natural killer-derived extracellular vesicles (NK-EVs) have a specific microRNA (miRNA) signature and are enriched in miRNAs related with Th1 functions. (**A**) Heatmap showing small RNA sequencing analysis of miRNAs differentially expressed in resting and activated human NK cells and their secreted EVs. Data are from NK cells isolated from five healthy donors. Significance was assessed using the Benjamini-Hochberg procedure and only miRNAs with an adjusted p-value < 0.05 are shown. (**B**) Histogram plot showing logarithmic fold-increase expression between activated NK cells and

*Figure 1 continued*

their released EVs. Only fold-changes with adjusted p-value < 0.05 and log fold-change > 2 are represented. MiRNAs significantly more expressed in cells than in their secreted exosomes are shown in green, while miRNAs significantly more represented in the EV fraction are shown in red. (**C**) Summary of molecular and cellular functions targeted by NK-EV miRNAs identified by unbiased ingenuity pathway analysis (IPA). The numbers indicate the molecules targeted by miRNAs over-represented in NK-EVs compared to NK cells. (**D**) Log fold-change in the small RNASeq expression of miRNAs significantly overexpressed in NK-EVs compared to their secreting cells, related to Th1 functions. Significance was assessed using Benjamini-Hochberg adjusted p-values; *p<0.05,**p<0.01, ***p<0.001, ****p<0.0001. (**E,F**) Quantitative real-time PCR of NK-EV upregulated miRNAs in resting NK cells, activated NK cells, and secreted EVs. Bars represent the mean ± SEM of cells and vesicles obtained from five healthy donors, normalized to the small nucleolar *RNU5G*. Data show the validation of CD4$^+$ T cell (**E**) and dendritic cell (DC) (**F**) function-related miRNAs, obtained by the 2-ΔΔCt method, using Biogazelle software. Significance was assessed by paired t-test; *p<0.05.

The online version of this article includes the following source data and figure supplement(s) for figure 1:

**Figure supplement 1.** Natural killer-derived extracellular vesicle (NK-EV) isolation and characterization.

**Figure supplement 1—source data 1.** Uncropped Western blots from the figure.

**Figure supplement 1—source data 2.** Original Western blot image showing CD63 expression.

**Figure supplement 1—source data 3.** Original Western blot image showing Tsg101 expression.

**Figure supplement 1—source data 4.** Original Western blot image showing Tsg101 expression.

**Figure supplement 2.** Natural killer-derived extracellular vesicle (NK-EV) microRNA (miRNA) content.

with the NK-EVs they release, we detected increased levels of cytosylated miRNA reads (positions 0, +1), and a decrease in guanosine additions (positions –1, 0). Although other significant changes in miRNA PtMs were observed in the different groups under study, they are supported by fewer number of reads.

Given that the number of detected isomiRs are influenced by the level of expression of individual miRNAs, we next studied the fraction of reads with PtMs within individual miRNAs. Remarkably, by analysing the percentage of reads with PtMs for the miRNAs specifically enriched in NK-EVs (*Figure 2D*, right panel), we found a significantly higher proportion of isomiRs in NK-EVs than in their secreting-activated NK cells. This suggests that miRNA molecules with PtMs are more likely to be included into EVs. In fact, five out of six of the most enriched NK-EV miRNAs show a higher number of reads with PtMs in NK-EVs compared to activated NK cells (*Figure 2—figure supplement 2A*). We next analysed the specific nucleotide mono-additions in miRNAs preferentially targeted into NK-EVs (*Figure 2E*). We found that both adenylation and cytosylation of canonical miRNA sequences accounted for the observed accumulation of PtMs in the NK-EV fraction. The levels of guanosylated miRNAs were very low in all samples, although more represented in resting NK cells, while no significant differences were observed in the levels of uridine additions, among the samples analysed.

Altogether, accumulation of PtMs in miRNAs enriched in NK-EVs suggests an active PtMs-dependent mechanism of specific miRNA sorting into EVs. However, the accumulation of modified miRNAs in NK-EVs was not observed in the case of the most expressed NK-EV miRNAs (*Figure 2—figure supplement 2B*), indicating additional layers of complexity in NK-EV-miRNA-specific packaging, for example passive sorting of the most abundant miRNAs.

Specific short sequence miRNA motifs are over-represented in EVs (*Figure 2F*). Binding of these motifs to RNA binding proteins, for example heterogeneous nuclear ribonucleoprotein A2/B1 (hnRNPA2B1), promotes miRNA sorting into EVs (*Villarroya-Beltri et al., 2013*). Bioinformatics sequence analyses showed over-represented short motifs in resting NK cells (UGCUG, *Figure 2—figure supplement 2C*), activated NK cells (UGCA, *Figure 2—figure supplement 2D*), and released NK-EVs (GGCAGU and UGGA, *Figure 2F*), respectively. Interestingly, the short EV-associated motif identified in T cell-derived small EVs GGAG was over-represented in NK-EV-miRNAs, and the tetra-nucleotide motif UGCA was also identified as a 'cell retention motif' in T cells, pointing towards similar mechanisms of miRNA sorting in lymphocytes(*Garcia-Martin et al., 2022*).

Our analyses of NK-EV-enriched specific miRNAs suggest that specific miRNAs packaging into NK-EVs may involve different mechanisms, including non-templated nucleotide additions to miRNA canonical sequences and short-motif miRNA recognition by RNA-binding proteins. Active targeting of specific miRNAs for EV release implies a putative role for these NK-EV-miRNAs in intercellular communication and regulatory functions.

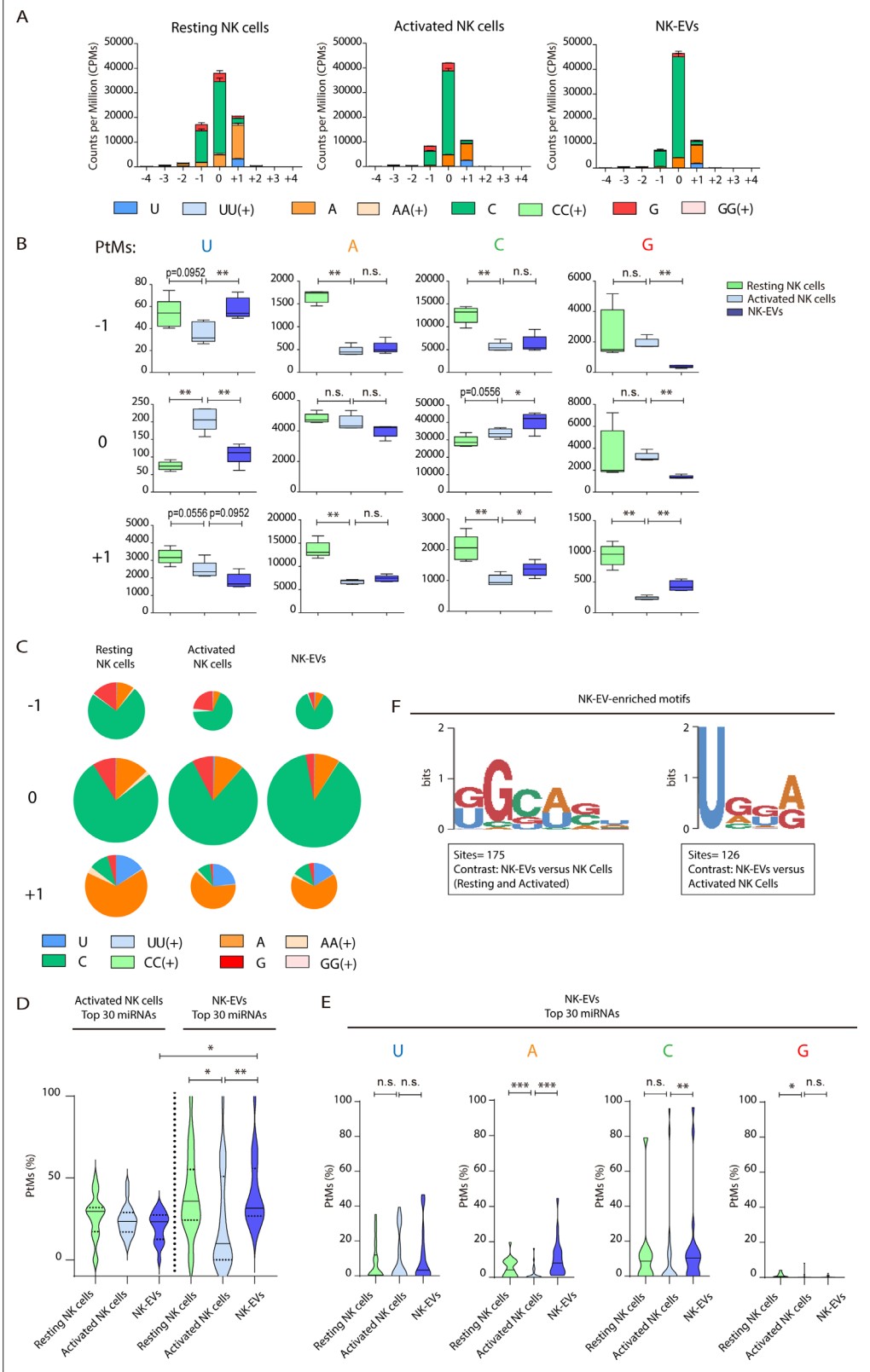

**Figure 2.** Resting natural killer (NK) cells, activated NK cells, and their released extracellular vesicles (EVs) contain microRNAs (miRNAs) with a different post-transcriptional modification (PtM) signature. (**A**) Bar charts showing the PtM profile of miRNAs from resting NK cells, activated NK cells, and NK-derived EVs (NK-EVs), expressed in normalized counts per million (CPMs) at the 3p-end. Modifications from the canonical sequence (either additions

*Figure 2 continued on next page*

*Figure 2 continued*

or substitutions) in positions ranging from –4 to +4 are represented. (**B**) Box and whiskers plots show the additions of U, A, C, and G at the indicated positions (ranging from –1 to +1) in miRNAs from resting NK cells, activated NK cells, and NK-EVs. Significance was assessed, comparing resting and activated NK cells and activated NK cells with their released small EVs using non-parametric t-test; *p<0.05, **p<0.01. (**C**) Pie charts show the proportion of the different modifications (U, A, C, and G) for each position (from –1 to +1) for resting NK cells, activated NK cells, and NK-EV miRNAs, respectively. The area of each pie chart is proportional to the total number of reads bearing the specified PtMs at the indicated positions. (**D**) Violin plots represent the fraction of miRNA reads with PtMs. Analysis was performed comparing the most differentially expressed miRNAs in activated NK cells compared to NK-EVs. The left panel summarizes the data obtained from the 30 miRNAs most expressed in activated NK cells, while the right panel shows the percentage of reads with PtMs from the 30 miRNAs with a higher expression in NK-EVs compared to their secreting activated NK cells. Individual miRNAs with less than 10 reads were excluded from the analysis. The median value is shown as a solid line and quartiles represented as dotted lines. Significance was assessed with the Kruskal-Wallis test; *p<0.05, **p<0.01. (**E**) Violin plots represent the fraction of miRNA reads with U, A, C, or G mono-additions at positions –1, 0, and +1. Analysis was performed for the 30 miRNAs more enriched in NK-EVs, comparing the PtM signature in resting cells and activated cells, and activated cells and their released NK-EVs. Significance was assessed with the non-parametric t-test Wilcoxon matched-pairs signed ranked test; *p<0.05, **p<0.01,***p<0.001. (**F**) Over-represented motifs in NK-EV-miRNAs, using the ZOOPS model. For each data set, all miRNAs annotated in miRBase 21 were used as background. Short motifs with adjusted E-value<0.2 are shown.

The online version of this article includes the following figure supplement(s) for figure 2:

**Figure supplement 1.** Resting natural killer (NK) cells, activated NK cells, and NK-derived extracellular vesicle (NK-EV) microRNAs (miRNAs) have a different isomiR signature.

**Figure supplement 2.** Post-transcriptional modification (PtM) characterization of the most expressed microRNAs (miRNAs) in resting and activated natural killer (NK) cells over-represented motifs.

**Figure supplement 3.** Natural killer-derived extracellular vesicle (NK-EV)-enriched microRNAs (miRNAs).

## NK-EVs promote Th1-like responses with increased T-bet expression and IFN-γ and IL-2 release

To dissect the putative immunomodulatory effects of shuttled NK-EV miRNAs, analyses of the individual miRNAs, either more abundant in the NK-EV fraction (*Figure 2—figure supplement 3A*) or more differentially enriched in NK-EVs as compared to activated NK producing cells (*Figure 2—figure supplement 3B*), and their in silico-predicted mRNA targets (*Supplementary file 2*) were carried out. As previously pointed out, miRNAs with key roles in immune function regulation, in particular with T cell responses, such as miR-10b-5p, miR-155-5p, or miR-92a-3p were highly expressed in NK-EVs.

To unravel the effects of NK-EVs on T cell function, CD4$^+$ T cells were isolated from human buffy coats and cultured either in the presence or absence of purified NK-EVs. An initial profiling of the cytokines secreted by NK-EV-treated CD4$^+$ T cells in non-cytokine polarizing conditions showed high levels of Th1 function-related cytokines, including IFN-γ and IL-2 (*Figure 3A and B*; *Figure 3—figure supplement 1A*). Unbiased IPA had identified cell death and survival as pathways potentially affected by NK-EV-enriched miRNAs, hence we evaluated CD4$^+$ T cell survival (*Figure 3—figure supplement 1B, C*). We found no significant differences in overall CD4$^+$ T cell survival after NK-EV treatment.

Further in vitro studies were performed to address the effects of NK-EV addition in non-polarizing and Th1-skewing conditions using a mixture of IL-2 and IL-12. Cultured CD4$^+$ T cells were analysed after 3 and 6 days, respectively. Flow cytometry analysis of cultured CD4$^+$ T cells showed an increase in the percentage of IFN-γ-producing T cells in the presence of NK-EVs in non-polarizing conditions. However, the effects were not evident in Th1-polarizing conditions (*Figure 3A*). Sandwich-ELISA analysis of cultured cells' supernatants revealed an increase in the levels of soluble IFN-γ in NK-EV-treated T cells in a non-cytokine-polarizing milieu (*Figure 3B*). In addition, incubation with NK-EVs also induced T-BET expression in non-cytokine-polarized T cells (*Figure 3C and D* and *Figure 3—figure supplement 1D*), correlating with a downregulation of the NK-EV miRNA target *GATA3* mRNA levels at day 3, which acts as a *TBX21* suppressor (*Figure 3D,E*; *Gagliani and Huber, 2017*). The increase of T-BET protein levels and downregulation of GATA3 were confirmed by flow cytometry analyses after incubation with NK-EVs (*Figure 3—figure supplement 1E*), although very low levels of GATA3 were detected. The increase of the NK-EV identified miRNAs miR-10b-5p and miR-92a-3p, which

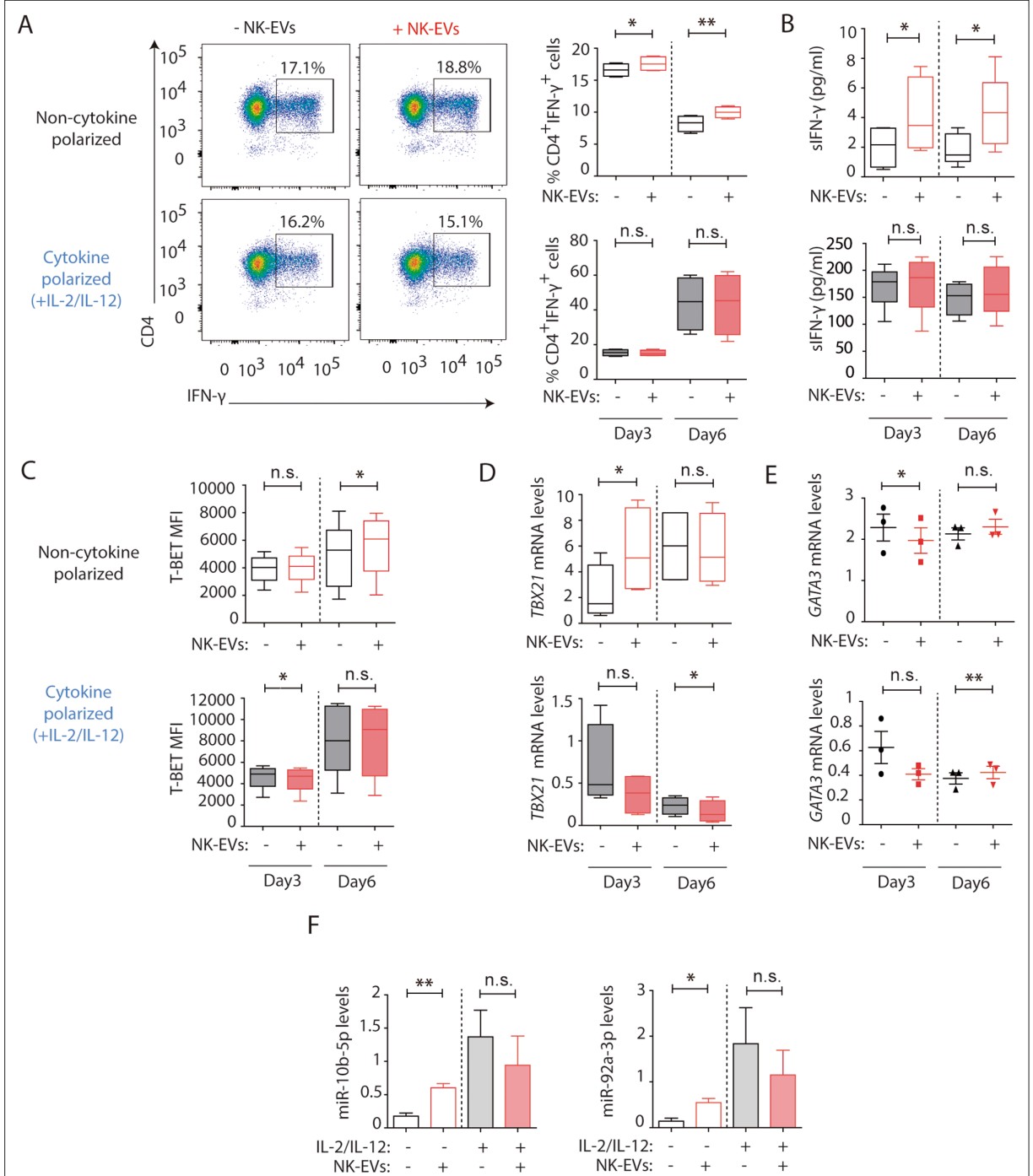

**Figure 3.** Natural killer-derived extracellular vesicles (NK-EVs) promote Th1 differentiation via *Gata3* downmodulation and *T-bet* de-repression correlating with increased levels of miR-10b-5p and miR-92a-3p. CD4+ T cells isolated from healthy human buffy coats were cultured either in non-polarizing or in cytokine Th1-polarizing conditions, with a mixture of IL-2 and IL-12, in the presence or absence of NK-EVs. (**A**) Flow cytometry analysis of isolated CD4+ T lymphocytes incubated under non-cytokine polarizing (upper panels) and Th1 cytokine-polarizing (with a mixture of IL-2 and IL-12, lower panels) conditions. A representative experiment is shown. Box and whiskers plots show the expression of CD4 and IFN-γ in gated live cells (min to max and median values), after addition of NK-EVs(Right panel). Plots show the quantification of n≥4 independent experiments. Significance was assessed with paired Student's t-test; *p<0.05, **p<0.01. (**B**) ELISA quantification of soluble IFN-γ in supernatants from cultured cells in the indicated conditions (unpolarized, upper panel; cytokine polarized, lower panel). The chart shows the median concentration from n≥4 independent experiments. Significance was assessed with paired Student's t-test; *p<0.05. (**C**) Flow cytometry analysis of isolated CD4+ T cells cultured in the different conditions. Graphs show the mean fluorescence intensity of T-BET in gated live single CD4+ T cells after 3 and 6 days of culture, respectively. Significance was assessed with paired Student's t-test; *p<0.05. (**D,E**) Quantitative real-time PCR at days 3 and 6 showing mRNA levels of *TBX21* (**D**) and *GATA3* (**E**),

*Figure 3 continued on next page*

*Figure 3 continued*

respectively, normalized to *GAPDH* and *ACTB*. Significance was assessed by paired Student's t test; *p<0.05. (**F**) Quantitative real-time PCR at day 3 to detect microRNA (miRNA) levels in CD4⁺ T cells after culture in the indicated conditions. MiR-10b-5p (left panel) and miR-92a-3p (right panel) relative expression is shown, and normalized to *RNU1A1* and *RNU5G*. Significance was assessed by paired Student's t test; *p<0.05.

The online version of this article includes the following source data and figure supplement(s) for figure 3:

**Figure supplement 1.** Natural killer-derived extracellular vesicle (NK-EV) microRNAs (miRNAs) and T cell function.

**Figure supplement 1—source data 1.** Uncropped Western blots from the figure.

**Figure supplement 1—source data 2.** Original Western blot image showing p150 expression.

**Figure supplement 1—source data 3.** Original Western blot image showing T-BET expression.

**Figure supplement 1—source data 4.** Original Western blot image showing GATA3 expression.

**Figure supplement 2.** Extracellular vesicle (EV) uptake blockade using Dynasore and size-exclusion chromatography (SEC) analyses confirm the specificity of natural killer-derived EV (NK-EV) microRNAs (miRNAs) effects.

**Figure supplement 3.** HEK and Raji-derived small extracellular vesicles (EVs) exhibit a distinct effect on T cell function than natural killer-derived EVs (NK-EVs).

have crucial functions in T cell responses (*Supplementary file 2*), was confirmed by RT-qPCR in non-polarized culture conditions but not in cytokine Th1-polarizing cultures, after 3 days of incubation with NK-EVs (*Figure 3F*). Moreover, to confirm the involvement of NK-EV-miRNAs in GATA3 down-regulation and subsequent T-bet de-repression and Th1 reprogramming in T cells, gold nanoparticles (AuNPs) bearing a combination of miR-10b-5p and miR-92a-3p, that specifically target this molecule, were synthesized and incubated with isolated CD4⁺ T cells (*Figure 3—figure supplement 1F*). Three days after incubation with miRNA-bearing nanoparticles, the effects of NK-EVs in T-BET and GATA3 expression were partially recapitulated, indicating the involvement of these miRNAs in the observed phenotype.

Since ultracentrifugation of cell culture supernatants does not allow to fully rule out the involvement of other molecules in the activated NK secretome that may account for the observed effects, for example effector proteins including IFN-γ (*Choi et al., 2020*), several additional experiments were performed. NK-EV T cell uptake blockade using Dynasore abolished NK-EV-driven increase of IFN-γ expression (*Figure 3—figure supplement 2A,B*), indicating the involvement of EV-uptake in promoting Th1-like responses. Moreover, small EVs were isolated from activated NK cells using size-exclusion chromatography (SEC), to exclude the contamination of protein complexes, lipoparticles, etc. Briefly, supernatants were added onto a chromatography column and fractions recovered and analysed for small EV markers expression. Fractions containing the CD81 and Tsg101 exosomal markers and lacking most protein aggregates were pooled (*Figure 3—figure supplement 2C,D*), and the effects of SEC-isolated NK-EVs on T cells were addressed, as above. Despite a significant loss of NK-EV material during the SEC purification steps (*Figure 3—figure supplement 2E,F*), confirmed by a reduction of the exosomal markers CD81 and Tsg101 to around 20% of the levels obtained by ultracentrifugation, a similar impact for SEC-isolated NK-derived small EVs in IFN-γ expression was observed (*Figure 3—figure supplement 2G,H*).

Finally, to rule out the non-specific effects of isolated EVs on T cell function, small EVs were isolated from two distinct human cell lines (namely the HEK-293, human epithelial kidney cells, and the Raji B lymphoblast cell lines), using our standard ultracentrifugation procedure. These vesicles showed no impact on IFN-γ secretion, that even decreased 6 days after incubation with Raji B cell-derived EVs, indicating the specificity of NK-EVs in skewing Th1 responses (*Figure 3—figure supplement 3*).

The activation of CD4⁺ T cells by NK-EVs was also evaluated. NK-EVs promoted T cell activation, as revealed by increased CD25 expression in non-polarizing conditions; whereas these effects were not further increased in a Th1 cytokine-polarized milieu (*Figure 4A*). Increased IL-2 release upon incubation with NK-EVs was confirmed by ELISA, 3 days after culture (*Figure 4B*). Since CD25 is also a hallmark of regulatory T cells (Treg), we next analysed the expression of *FOXP3* and *IL10* production in NK-EV-treated T lymphocytes (*Figure 4C*). Although no significant differences were found, increased *FOXP3* levels (p-value < 0.1) in non-polarized NK-EV-treated cells were observed compared to non-EV-treated cells. However, *IL10* secretion was not affected by NK-EV treatment, and was even reduced 3 days after culture in polarizing conditions. These data suggest that additional levels of regulation

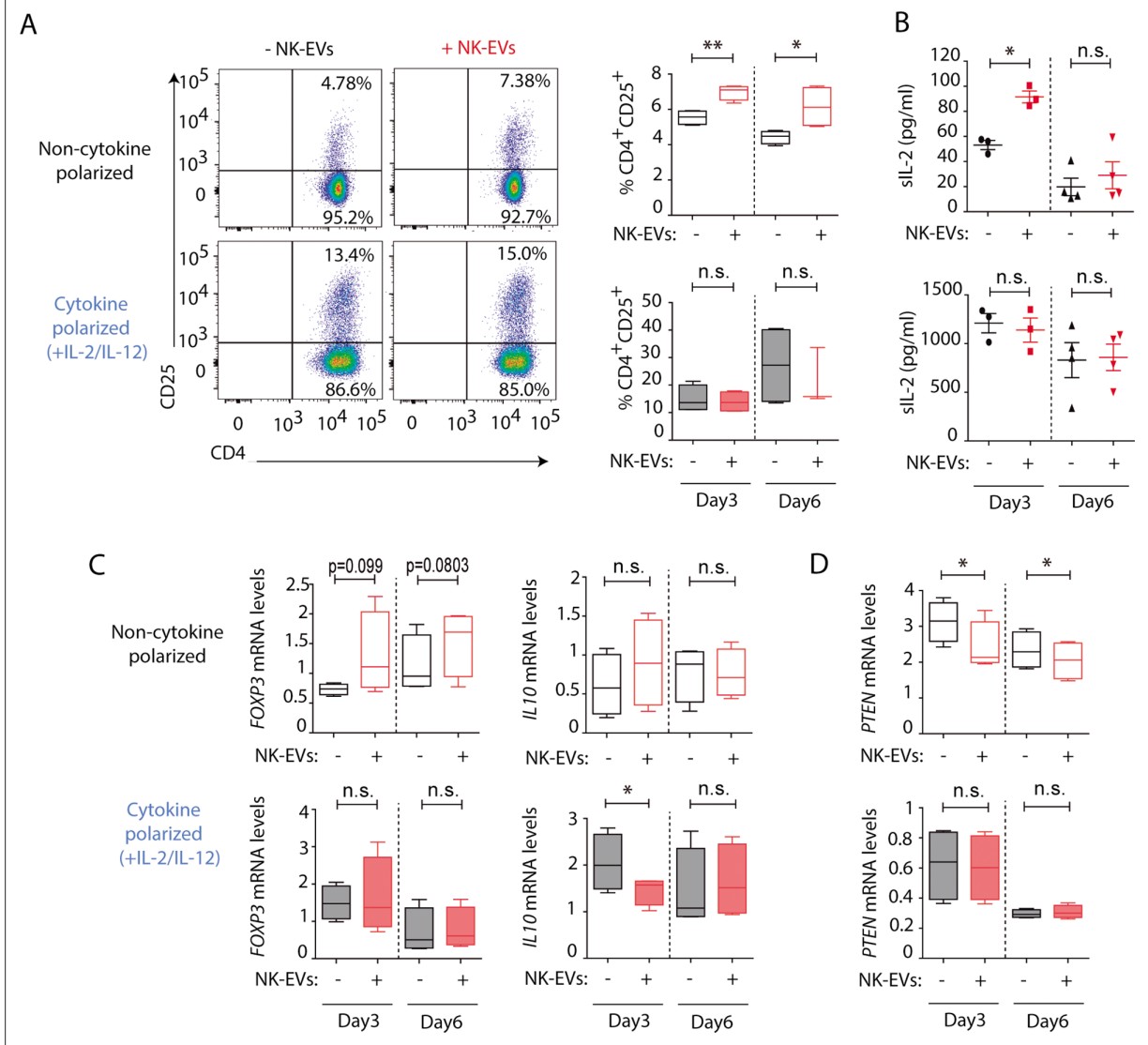

**Figure 4.** Natural killer-derived extracellular vesicles (NK-EVs) promote CD4+ T cell activation and IL-2 release but not regulatory T cell (Treg) responses. (**A**) Flow cytometry analysis of isolated CD4+ T cells incubated under non-cytokine-polarizing (upper panel) and Th1 cytokine-polarizing (with a mixture of IL-2 and IL-12, lower panel) conditions. Dot plots show the expression of CD4 and CD25 in gated single live cells ± SEM, after addition of NK-EVs. Plots show the quantification of n≥4 independent experiments. Significance was assessed with paired Student's t-test; *p<0.05, **p<0.01. (**B**) ELISA quantification of soluble IL-2 in supernatants from CD4+ cultured T cells in the indicated conditions (unpolarized, upper panel; cytokine polarized, lower panel). The graph shows the mean concentration from n≥3 independent experiments. Significance was assessed by paired Student's t-test; *p<0.05. (**C,D**) Quantitative real-time PCR showing *FOXP3* (**C**, left); *IL10* (**D**, right) and *PTEN* (**D**) mRNA levels at days 3 and 6 in CD4+ T cells after culture in the indicated conditions. Relative expression is shown, normalized to *GAPDH* and *ACTB*. Significance was assessed by paired Student's t test; *p<0.05.

The online version of this article includes the following figure supplement(s) for figure 4:

**Figure supplement 1.** mRNA target modulation in CD4+ T cells mediated by natural killer-derived extracellular vesicles (NK-EVs).

**Figure supplement 2.** T cell activation is specific of natural killer-derived extracellular vesicles (NK-EVs).

participate in T cell responses, and that NK-EVs may limit suppressive Treg functions in Th1-skewing conditions.

Analysis of additional putative NK-EV miRNA targets described in *Supplementary file 2* was carried out (*Figure 4D* and *Figure 4—figure supplement 1*). Functional regulatory molecules were downregulated by NK-EVs, including *PTEN* (*Figure 4D*), which is involved in T cell homeostasis; *IL6*, which was reduced after NK-EV incubation (p-value <0.1); *TGFB* and INPP5D gene that encodes the SHIP1 protein (*Figure 4—figure supplement 1A, D, F*). On the contrary, the pleiotropic *TNFA*

cytokine was upregulated shortly after NK-EV incubation in non-polarizing conditions, but decreased in Th1 cytokine-polarizing conditions (*Figure 4—figure supplement 1E*). The specific involvement of NK-EVs in T cell activation was further confirmed after dynamin-dependent endocytosis blockade using Dynasore (*Figure 4—figure supplement 2A,B*) and reproduced with SEC-purified NK-EVs, despite the loss of material with this procedure (*Figure 4—figure supplement 2C,D*). Also, incubation of T cells with either HEK or Raji-derived small EVs failed to reproduce NK-EV effects, confirming the specificity of the observed phenotype (*Figure 4—figure supplement 2E,F*). Interestingly, Raji-derived small EVs promoted a reduction in CD25 expression, indicating that NK-EV effects on T cells are not driven by small EVs from any lymphocyte.

Altogether our data indicate a complex regulation of T cells responses mediated by NK-EVs.

## NK-EVs promote DC-mediated Th1 responses, enhancing monocytes and moDC presentation and costimulatory functions

Dendritic cells are critical for the establishment of effective T cell responses (*Sallusto and Lanzavecchia, 2002*). To ascertain the role of NK-EVs in DC function, human monocytes were obtained, and cultured either in DC-polarizing conditions (supplemented with a mixture of GM-CSF and IL-4) or without cytokines (*Sallusto and Lanzavecchia, 1994*). The effects of the addition of NK-EVs were assessed in both experimental conditions (monocytes and monocyte-derived DCs [moDCs]). Flow cytometry analyses revealed that, although no significant differences were observed in cell survival and apoptosis (*Figure 5—figure supplement 1A,B*), the presentation and costimulatory capacities of DCs were enhanced by NK-EVs exposure (*Figure 5A–E*). Indeed, both NK-EV-treated monocytes and moDCs showed increased levels of MHC-II (*Figure 5A*) and the costimulatory CD86 molecule (*Figure 5B*). Additionally, non-cytokine-polarized monocytes exhibited a higher DC polarization, marked by an increased percentage of CD11c⁺DC-SIGN⁺ cells after incubation with NK-EVs (*Figure 5C*). Interestingly, NK-EVs' addition correlated with increased levels of IFN-γ and IL-12 secretion in monocytes, but not in moDCs that exhibited very low levels of these cytokines (*Figure 5D and E*).

Several putative targets for NK-EV miRNAs with important functions in DCs (*Supplementary file 2*) were analysed after treatment with NK-EVs, including *KLF13*, *FOXO1*, *NOTCH1*, and *VEGFR2*, but no significant expression differences were found (*Figure 5—figure supplement 1C-F*).

## NK-EV miRNAs partially recapitulate NK-EV-polarizing functions

To address whether miRNAs contained in NK-EVs may account for the observed Th1 immune deviation, the effects of in vivo footpad administration of nanoparticles carrying the NK-EV-miRNAs miR-10b-5p, miR-92a-3p, and miR-155-5p (*Figure 6—figure supplement 2A*) were analysed. Interestingly, splenocytes isolated from nanoparticle-treated mice activated in vitro with anti-CD3 and anti-CD28 antibodies showed increased T cell activation after miRNA-bearing treatment compared to control nanoparticles, assessed by increased CD25 expression in CD4⁺ lymphocytes (*Figure 6A and B*). qPCR analysis of total splenocytes (*Figure 6C*) and popliteal draining lymph nodes (*Figure 6D*) showed increased mRNA levels of the Th1 hallmark *Ifng* after footpad administration of NK-EV miRNAs, in particular with miR-155 and a mixture of all three NK-EV miRNAs. Increased secretion of IFN-γ was also confirmed in supernatants from in vitro-activated splenocytes by sandwich-ELISA (*Figure 6E*). Although no significant differences were found in the percentage of IFN-γ expressing CD4⁺ T cells (*Figure 6F*, lower panel), we could observe a similar tendency in T cells from mice treated with miR-10b-5p and miR-92a-3p, which exhibited a significantly enhanced expression of this cytokine upon TCR-stimulation, compared to non-miRNA-treated cells (*Figure 6F*, upper panel). The effects of miR-10b-5p, miR-92a-3p, and miR-155-5p delivery in CD8⁺ T cells IFN-γ secretion of this cytokine were more robust (*Figure 6G*). The levels of *Tbx21* were slightly increased in miRNA-treated mice (*Figure 6—figure supplement 1A*), although the differences were not significant, likely due to the kinetics of T-bet upregulation after Th1 commitment (*Szabo et al., 2000*). Importantly, as observed in the experiments after NK-EV treatment in vitro, no differences were observed in the Treg compartment upon nanoparticle-based miRNA delivery, as addressed by the percentage of CD4⁺CD25⁺Foxp3⁺ cells (*Figure 6—figure supplement 1B*) and the levels of *Il10*, neither in the spleen (*Figure 6—figure supplement 1C*), nor in the popliteal draining lymph nodes (*Figure 6—figure supplement 1D*). ELISA analysis of supernatants splenocytes from treated mice showed no significant differences in the levels of IL-2 secretion (*Figure 6—figure supplement 1E*). Low levels of both IFN-γ

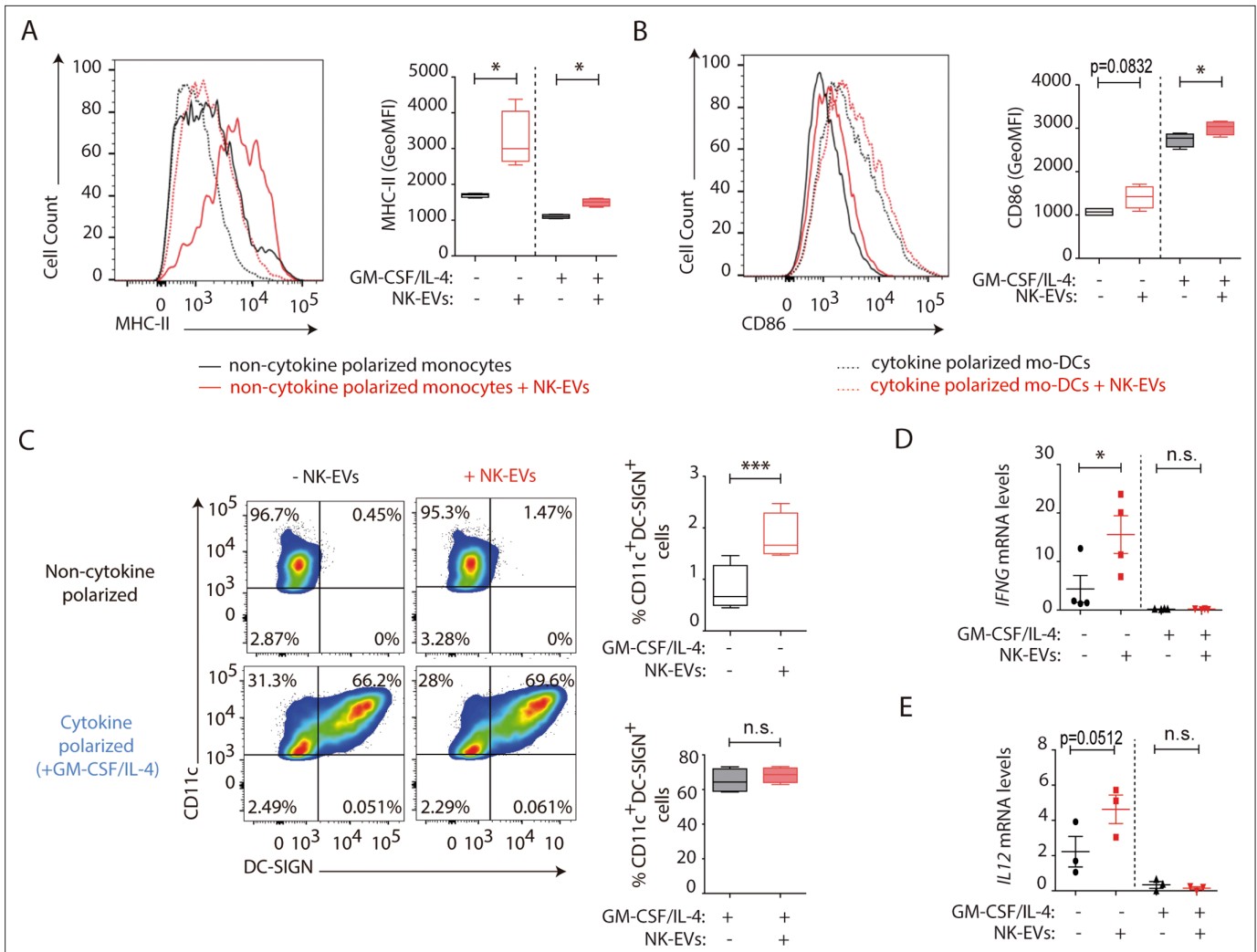

**Figure 5.** Natural killer-derived extracellular vesicles (NK-EVs) enhance monocyte and monocyte-derived dendritic cells (moDCs) polarization, costimulatory and presentation activities. CD14+ monocytes isolated from buffy coats from healthy donors were cultured under non-polarizing and cytokine monocyte-derived DC-polarizing conditions, with a mixture of GM-CSF and IL-4, with or without NK-EVs. (**A,B**) Flow cytometry analysis of isolated CD14+ monocytes in the different culture conditions. Histogram plots show the expression of MHC-II (**A**) and CD86 (**B**), after 6 days of culture in the indicated conditions, either in the presence (dotted line) or absence (solid line) of polarizing cytokines. A representative plot of n≥4 experiments of the mean fluorescence intensity (MFI) of these proteins in live cultured cells (left panel), and their quantification (right panel) are shown. Significance was assessed by paired Student's t test; *p<0.05. (**C**) Flow cytometry analysis of isolated CD14+ monocytes in the different culture conditions. Dot plots show the expression of CD11c and DC-SIGN in the indicated culture conditions (left panel) and their quantification (right panel). Significance was assessed by paired Student's t test; ***p<0.001. (**D,E**) Quantitative real-time PCR showing *IFNG* (**D**) and *IL12* (**E**) mRNA levels in CD14+ cells 6 days after culture in the indicated conditions. Relative expression is shown, normalized to *GAPDH* and *ACTB*. Significance was assessed by paired Student's t test; *p<0.05.

The online version of this article includes the following figure supplement(s) for figure 5:

**Figure supplement 1.** Natural killer-derived extracellular vesicles (NK-EVs) impact on monocyte and monocyte-derived dendritic cell (moDC) function.

and IL-2 cytokines were detected in serum from treated mice, however miR-10b-5p and miR-155-5p bearing nanoparticles promoted a slight increase of these cytokines (*Figure 6—figure supplement 1F,G*). To further confirm the specificity of NK-EV-enriched miRNAs in promoting Th1 polarization and activation, AuNPs bearing three miRNAs preferentially excluded from the NK-EV fraction, namely hsa-miR-124, hsa-miR-3667, and hsa-miR-4518 were generated (*Figure 6—figure supplement 2B*), and their effects after footpad administration in immunocompetent C57BL/6 mice were evaluated, as for NK-EV miRNAs. These experiments showed no effects of these nanoparticles, as observed for NK-EV-enriched miRNAs. Neither activation nor IFN-γ secretion (*Figure 6H*) were affected by non-NK-EV miRNAs, further confirming the specificity of NK-EV miRNAs in the observed phenotype.

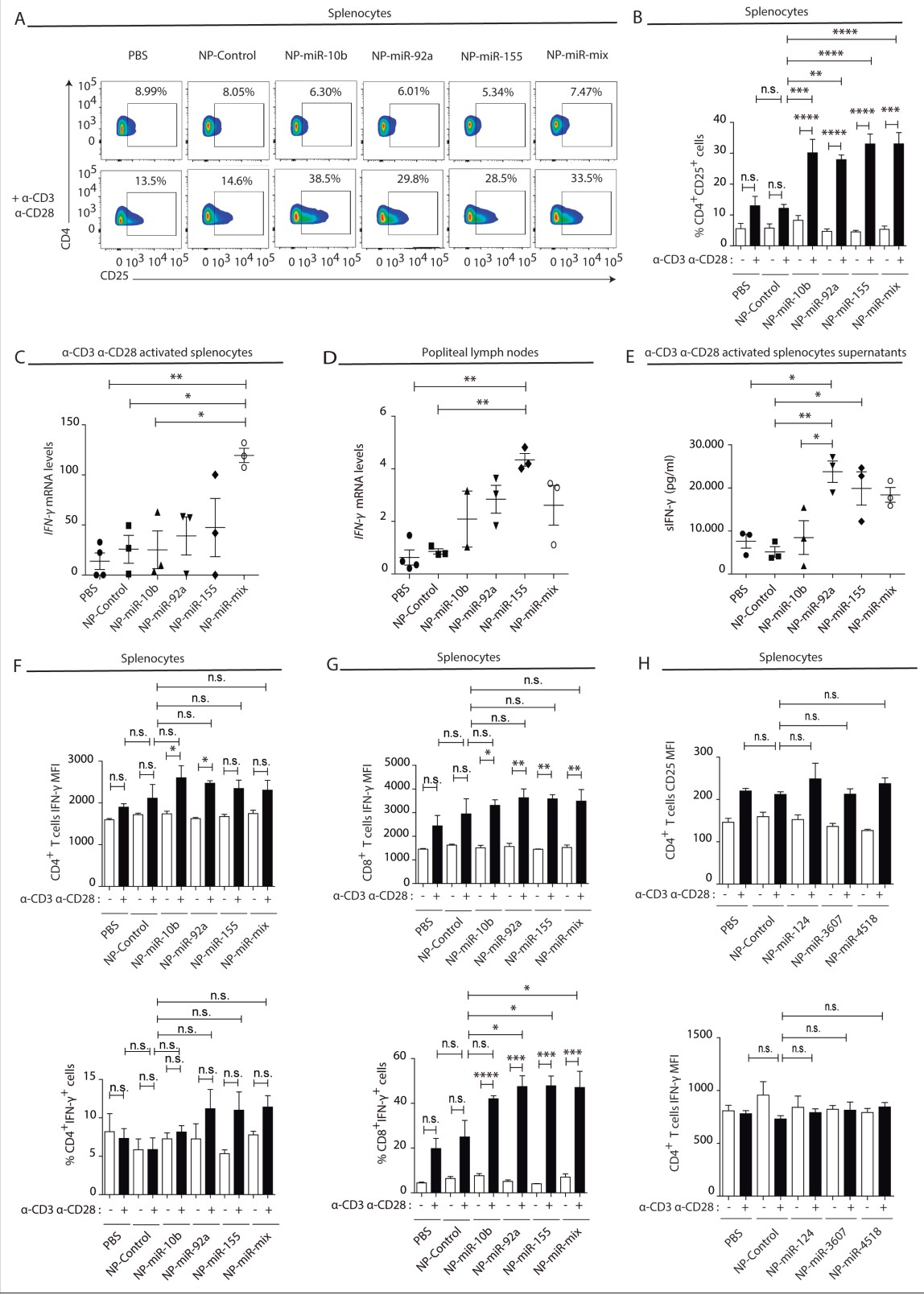

**Figure 6.** Natural killer-derived extracellular vesicle (NK-EV) T cell function-related microRNAs (miRNAs) partially mimic NK-EV-mediated effects in vivo. Nanoparticles (NPs) bearing the identified NK-EV miRNAs (miR-10b, miR-92a, and miR-155) were generated and their functional effects were evaluated in vivo after footpad injection of control and miRNA-loaded NPs in wild-type C57BL/6 mice. (**A**) Flow cytometry analysis of live splenocytes isolated 6 days after footpad injection with the indicated control or miRNA-loaded NPs. Isolated cells were either left unstimulated (upper panel) or incubated

*Figure 6 continued on next page*

*Figure 6 continued*

for 16 hr with anti-CD3 and anti-CD28 antibodies (lower panel) before analysis. Density plots show the expression of CD4 and CD25 in TCRβ$^+$CD4$^+$-gated live T lymphocytes. A representative plot of n≥3 mice is shown. (**B**) Bars summarize the quantification of the percentage of CD4$^+$CD25$^+$T cells shown in (**A**) from n≥3 mice in untreated splenocytes (white) and anti-CD3 plus anti-CD28 stimulated splenocytes (black). Significance was assessed by two-way ANOVA, followed by Tukey's test; **p<0.01, ***p<0.001, ****p<0.0001. (**C,D**) Quantitative real-time PCR showing *Ifng* mRNA levels in anti-CD3 and anti-CD28-activated splenocytes (**C**) and draining popliteus lymph nodes (**D**), 6 days after footpad nanoparticle-based miRNA delivery. Relative expression is shown, normalized to *Gapdh* and *Actb*. Significance was assessed by one-way ANOVA Bonferroni test; *p<0.05, **p<0.01. (**E**) ELISA quantification of soluble IFN-γ in supernatants from splenocytes harvested and isolated 6 days after miRNA delivery and cultured with anti-CD3 and anti-CD28 antibodies for 16 hr. The graph shows the mean concentration from n≥3 independent experiments. Significance was assessed by one-way ANOVA Bonferroni test; *p<0.05, **p<0.01. (**F,G**). Upper panel bar charts show the mean fluorescence intensity (MFI) expression of IFN-γ and lower panels the percentage of IFN-γ expressing cells, analysed by flow cytometry in CD4$^+$ T cells (**F**), CD8$^+$ T cells (**G**). Splenocytes were either left unstimulated (white) or stimulated with anti-CD3 plus anti-CD28 antibodies for 16 hr (black). Significance was assessed by two-way ANOVA, followed by Tukey's test; *p<0.05,**p<0.01, ***p<0.001, ****p<0.0001. (**H**) The effects of the non-NK-EV-associated control miRNAs (hsa-miR-124, hsa-miR-3607, and hsa-miR-4518) were also evaluated in vivo after footpad injection of control and miRNA-loaded NPs in immunocompetent wild-type C57BL/6 mice. Bar charts show the MFI expression of CD25 (upper panel) IFN-γ (lower panel), respectively, analysed by flow cytometry in CD4$^+$ T cells. Significance was assessed by two-way ANOVA, followed by Tukey's test.

The online version of this article includes the following figure supplement(s) for figure 6:

**Figure supplement 1.** Natural killer-derived extracellular vesicle (NK-EV) microRNAs (miRNAs) partially mimic NK-EV-mediated effects in vivo in spleen and draining popliteal lymph nodes.

**Figure supplement 2.** Natural killer-derived extracellular vesicle (NK-EV) microRNAs (miRNAs) and non-NK-EV miRNA candidates loaded into gold nanoparticles.

In summary, activated NK cells release a specific set of EV-miRNAs, likely by mechanisms involving PtMs and specific-short miRNA motifs recognition, which are related with Th cells effector functions and may therefore contribute to orchestrate the immune response (*Figure 7*).

## Discussion

In recent years, the potential of EVs as tools for the treatment, diagnosis, and prognosis of several diseases has been intensively studied. Most studies to date suggest that EV miRNA cargo can be exploited to specifically target pathological recipient cells, including immune cells.

NK cells play a pivotal role in the recognition and killing of pathological cells, and recent evidence supports the potential of NK-secreted EVs for therapy (*Del Vecchio et al., 2021*). Several reports have described the modulation of NK cell effector functions by EVs, including exosomes. Tumour-derived EVs exert immune suppressive functions, dampening lymphocyte cytotoxicity (*Ashiru et al., 2010*; *Fernández-Messina et al., 2010*; *Sharma et al., 2018*). Studies aimed at characterizing the biological effect of DC-derived EVs have indicated that they can boost NK and T cell effector functions (*Pitt et al., 2014*; *Sobo-Vujanovic et al., 2014*; *Munich et al., 2012*; *Radomski et al., 2019*). In fact, DC-derived exosomes capabilities are being used in clinical trials for immune maintenance after chemotherapy (*Besse et al., 2016*) and cancer immunotherapy (*Fernández-Delgado et al., 2020*).

NK-EVs can kill target cells directly (*Lugini et al., 2012*) due to the presence of cytotoxic proteins amongst the cargo, as shown using proteomics assays (*Wu et al., 2019*; *Korenevskii et al., 2018*). Recently, specific NK-EV miRNAs were shown to exert antitumoural effects in neuroblastoma (*Neviani et al., 2019*) and pancreatic cancer progression (*Sun et al., 2019*). In particular miR-186-5p identified as a downmodulated miRNA in patients with a high risk of developing neuroblastoma was found to inhibit key components of the TGF-β pathway limiting tumour immune evasion. However, the specific NK-EV miRNA repertoire and the impact of NK-EV-miRNAs in the immune response remains far from being fully elucidated.

In this study, we identify a specific miRNA signature of human NK cells (resting and activated) and their secreted EVs by small RNA NGS. Also, we show that a distinct pattern of PtMs is found in resting NK cells, activated NK cells, and NK-EVs. Global analysis of miRNA PtMs also identified a higher proportion of reads with PtMs for the miRNAs more highly expressed in the NK-EV fraction compared to both the relative levels of PtMs in these miRNAs at their parental cells, and to the percentage of PtMs found in NK-EVs of activated NK-cell-enriched miRNAs. Also, in this context, activated NK cells miRNAs exhibited significantly decreased levels of PtMs compared to resting cells. Indeed, several miRNAs, for example miR-122-5p, miR-409-3p, and miR-451a, are highly modified in resting NK cells,

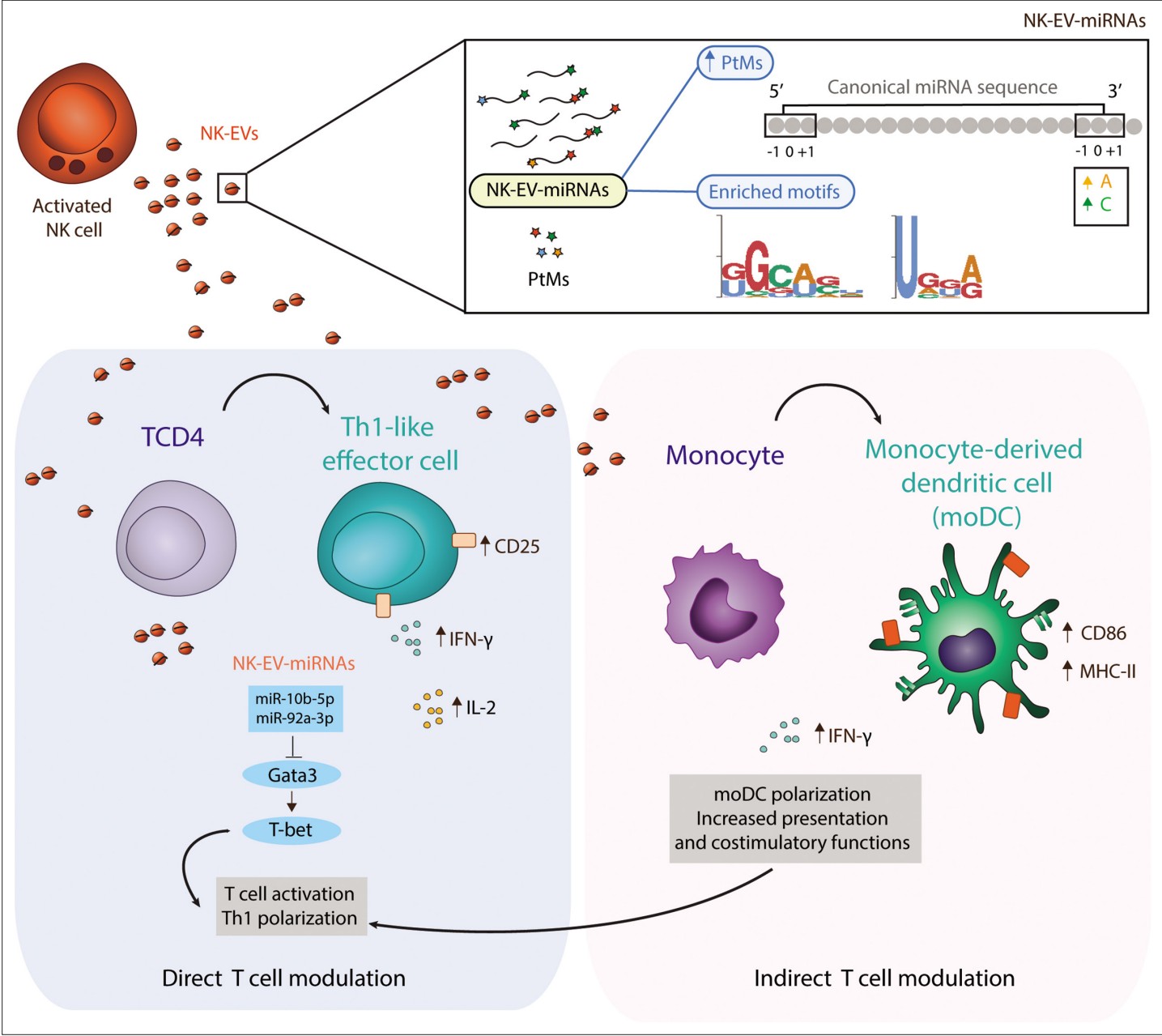

**Figure 7.** Natural killer-derived extracellular vesicle (NK-EV) microRNAs (miRNAs) promote Th1-like responses. NK cells release small EVs that are enriched in post-transcriptionally modified miRNAs and specific short motifs. NK-EVs have a direct impact on T cells fate and function, promoting Th1-like responses, marked by increased levels of IFN-γ and IL-2 release. NK-EV-miRNAs miR-10b-5p and miR-92a-3p may account, at least partially, for the observed effects, via GATA3 downmodulation and subsequent T-bet de-repression that drives Th1 skewing. Additionally, NK-EVs promote monocyte polarization to monocyte-derived dendritic cells (moDCs) and enhance their presentation and costimulatory capacities, by upregulating MHC-II and CD86, respectively. This pathway also may contribute to the regulation of Th1-like responses mediated by NK-EVs.

and in activated NK cell-derived EVs, while absent in activated NK cells. Thus, it is tempting to speculate that upon activation, post-transcriptionally modified NK-EV-miRNAs are preferentially targeted to small EVs. These results strongly suggest a PtM-dependent mechanism of miRNA-specific shuttling into NK- EVs.

Analyses of the specific nucleotide additions to the canonical miRNA sequences at different positions also highlighted a different signature for miRNAs in resting and activated NK cells and their released EVs. The post-transcriptional addition of non-templated nucleotides, particularly to the 3'ends of RNA molecules, is the most frequent and conserved RNA PtM. Interestingly, these modifications have an

impact on miRNA stability, and gene regulation, thus on its regulatory capacities (*Yu and Kim, 2020*). Furthermore, 3'uridylation is related to miRNA turnover during T cell activation (*Gutiérrez-Vázquez et al., 2017*). Non-templated addition of nucleotides has been differentially detected in diverse body fluids. PtMs of miRNAs have been previously reported to determine RNA fate and 3' adenylation has been found to be over-represented in cells, while uridylation in exosomes from B cells suggesting a PtM-dependent specific cargo of miRNAs into small EVs (*Koppers-Lalic et al., 2014*).

In our human NK cell model, we observed that the fraction of both adenylated and cytosylated miRNAs was abundant in resting NK cells and decreased upon activation. Remarkably, a significant enrichment of miRNAs with non-templated 3'-end additions of adenines and cytosines was found in the NK-EV fraction. It is therefore conceivable that cytosylated and adenylated miRNAs are preferentially sorted into small EVs upon NK cell activation.

Moreover, over-represented short sequence motifs were found in NK-EV-miRNAs. In particular, the previously identified GGAG exosomal-sorting motif in T cell-derived small EVs (*Villarroya-Beltri et al., 2013*) was also enriched in NK-EV-miRNAs, indicating that binding of sumoylated ribonucleoproteins may also be involved in the control of NK-EV-miRNAs sorting. Altogether, our data indicate that a complex regulation, including miRNA PtMs and short motif recognition patterns, may determine the inclusion of specific miRNAs with potential regulatory effects into NK-EVs.

Interestingly, NK-EVs are enriched in miRNAs with important functions in the regulation of immune responses, in particular related to Th1 responses. Other miRNAs with immunomodulatory functions are also abundantly expressed in NK cell-derived EVs, for example miR-20a-5p and miR-25-3p. These two miRNA are transferred through the immune synapse, with an impact on germinal centre reaction and antibody production (*Fernández-Messina et al., 2020*). Incubation of CD4+ T cells with NK-EVs induced an increase of IFN-γ secretion and T-bet expression. This effect correlated with increased levels of the NK-EV-associated miR-10b-5p and miR-92a-3p and reduced levels of the T-bet suppressor GATA3, which is a putative target for these miRNAs (*Yu and Kim, 2020*). Since NK-EVs isolated by ultracentrifugation may associate with other molecules from the secretome, such as IFN-γ, additional experiments blocking EV uptake and isolating NK-EVs by SEC confirmed that the observed effects were mediated by NK-EVs. Moreover, EVs from other cell types, including the lymphoblast B cell line Raji did not show any enhancing effect on T cell function, indicating the specificity of NK-EVs promoting the observed phenotype.

In addition, nanoparticle-based delivery of NK-EV-miRNAs miR-92a and miR-155 in vivo promoted IFN-γ production. Thus, NK-EV miRNAs miR-10b-5p and miR-92a-3p uptake may promote GATA3 downregulation and subsequent T-bet de-repression, reprogramming recipient T cells towards the Th1 phenotype. In addition, NK-EVs drive CD4+ T cell activation, marked by increased CD25 expression and IL-2 production, but do not significantly affect the Treg compartment. Indeed, although slightly increased levels of Foxp3 were found in T cells after incubation with NK-EVs, no increase in IL-10 secretion was observed and a reduction of this cytokine was found in polarizing conditions, suggesting that NK-EVs may be limiting Treg anti-inflammatory functions in these conditions. A similar trend was observed in vivo after nanoparticle delivery of NK-EV-specific miRNAs. In this regard, we detected reduced levels of the pro-inflammatory cytokine IL-6 and TGF-β in T cells upon NK-EV addition. However, the tumour necrosis factor (TNF)-α was found to be increased in T cells shortly after NK-EV addition, in agreement with its presence in the NK92 cell line-derived exosomes (*Zhu et al., 2017*). It is important to highlight that EVs do not only contain miRNAs, but also lipids and proteins that also may contribute to the immunomodulatory effects of the NK-EVs. The presence of these non-miRNA bioactive molecules in NK-EVs may account for some of the differences observed between the effects of synthetic miRNAs and NK-EVs.

Our data also show that NK-EVs may impact on T cell responses indirectly by increasing the polarization of monocytes to moDCs, and enhancing their presentation and costimulatory capacities, upregulating MHC-II and CD86, respectively. Hence, NK-EVs promote Th1 responses, both by directly targeting CD4+ T cells and by increasing APCs stimulating functions, necessary for the establishment of effective T cells responses. In this regard, many tumours dampen Th effector cell-mediated immunosurveillance, and many immunotherapeutic approaches have focused in restoring Th1 responses (*Knutson and Disis, 2005*). Immune therapy has revolutionized the treatment of cancer, however, long-lasting effects are only observed in about 15% of the patients. This might be due to the variety of cancer immune evasion mechanisms, including immune-suppressive mediators in the tumour

microenvironment, impairment of CTL responses, promotion of T cell tolerance, and/or exhaustion and polarization from Th1 to Th2, among others (*Marron et al., 2021*). Thus, we propose that NK-EV-mediated immune deviation towards Th1 may be exploited and taken into account when designing EV-based therapies. Interestingly, although primary human NK-EVs isolation may be tedious and time-consuming, a large-scale NK-EV isolation protocol has been set up and may help translate into clinical therapies (*Jong et al., 2017*).

In summary, here, we provide new insights on the miRNA composition and impact of NK-EVs on immune responses, promoting both directly and indirectly (via APCs) Th1 responses. Unravelling the effects and mechanisms underlying NK-EV-mediated immune targeting, in particular Th2 to Th1 immune deviation, may help to improve therapeutic strategies.

## Materials and methods

### In vitro cell culture, antibodies, and reagents

Primary human NK cells were obtained from healthy donor's buffy coats, using a protocol modified from *Roda-Navarro et al., 2006*. Briefly, PBMCs were isolated by centrifugation in Ficoll-hypaque and allowed to adhere for 30 min. Thereafter, resting NK cells were obtained using the human NK cell isolation kit (MACS Miltenyi Biotec), following the manufacturer's instructions, and lysed in QIAZOL (QIAGEN) buffer for RNA extraction and analysis. For activated NK cultures, PBMCs (seeded at 1.5 × 10$^6$ cells/well) were incubated for 5–6 days in RPMI medium (containing 2 mM L-glutamine, 1 mM sodium pyruvate, 0.1 mM non-essential amino acids, 100 U/ml penicillin-streptomycin [Biowest], 50 µM β-mercaptoethanol [Merck], 10 mM HEPES [Lonza]) supplemented with 10% human serum (HS), 10% foetal bovine serum (FBS), a mixture of irradiated feeder cells (721.221 and RPMI8866, at 0.5 × 10$^6$ per well) and a combination of NK-polarizing cytokines; 10 U/ml IL-12 (Peprotech) and 25 ng/ml IL-18 (MBL). Then, cells were allowed to proliferate for two additional days by activation with 50 U/ml IL-2 (Peprotech) and 10% HS and 10% FBS. NK cell-enriched cultures were analysed by flow cytometry to confirm the expansion of the NK cell population (CD3$^-$CD56$^+$). NK cells were purified using the NK isolation kit. Isolated NK cells were cultured for 72 hr in RPMI supplemented with 10% EV-depleted FBS and 50 U/ml IL-2 (by ultracentrifugation at 100,000× *g* for 16 hr) for EV accumulation.

The human cell lines HEK and Raji were maintained in DMEM and RPMI medium, respectively (containing 2 mM L-glutamine, 1 mM sodium pyruvate, 0.1 mM non-essential amino acids, 100 U/ml penicillin-streptomycin [Biowest], 10 mM HEPES [Lonza]) supplemented with 10% FBS. Cell lines were purchased from ATCC and tested negative for mycoplasma contamination.

Antibodies and primers are summarized in *Supplementary file 3*. Primers for specific mature miRNAs detection were purchased from EXIQON-QIAGEN.

### Animals

Wild-type C57BL/6 mice were housed in specific pathogen-free conditions according to European Commission recommendations at the Centro Nacional de Investigaciones Cardiovasculares (CNIC) animal facility in Madrid. Experiments were performed with male mice aged 10–12 weeks. All experimental methods and protocols were approved by the CNIC and the Comunidad Autónoma de Madrid and conformed to European Commission guidelines and regulations.

### NK-EV isolation and characterization

EVs were purified from supernatants after accumulating for 72 hr from 15 × 10$^6$ NK cells (2 × 10$^6$ cells/ml in RPMI supplemented with 10% small EV-free FBS and 50 U/ml IL-2), by serial centrifugation as previously described (*Fernández-Messina et al., 2010*). EVs were resuspended either in EXO-free RPMI for functional analysis, PBS for EV characterization by NanoSight (*Fernández-Messina et al., 2020*), or in Laemmli loading buffer for Western blot. For resting NK cell-derived EVs analysis, NK cells were isolated from healthy human buffy coats and the obtained resting NK cells (up to 5 × 10$^6$ cells) were allowed to secrete EVs for 72 hr in small EV-free medium before NK-EV isolation.

For small RNA sequencing and qPCR analysis, EV pellets obtained after 100,000× *g* ultracentrifugation of NK cell supernatants were lysed in QIAZOL (QIAGEN) and RNA was isolated using the miRNeasy Mini Kit, as described above. Culture medium pellet after 100,000× *g* ultracentrifugation

alone was analysed by SEC to rule out possible serum contaminations in NK-EV samples, as previously described (*Fernández-Messina et al., 2020*).

Additionally, size-exclusion (SEC) small EV isolation was performed using qEV/35 nm SEC columns (IZON), following the manufacturer's instructions. Collected fractions were analysed by dot blot for small EV markers expression and protein concentration was determined by Nanodrop. Fractions that included the exosomal markers Tsg101 and CD81 and low levels of protein aggregates were pooled and ultracentrifuged at 100,000× *g* with PBS before functional analyses.

All the relevant data of our experiments were submitted to the EV-TRACK knowledgebase (EV-TRACK ID: EV210234) (*Van Deun et al., 2017*).

## Electron microscopy

Isolated NK-EVs were analysed by electron microscopy (EM). Briefly, isolated NK-EV pellets were resuspended and fixed in PBS containing 2% PFA. Briefly, EM analyses was performed by floating a Formvar-carbon EM grid on top of one drop of freshly prepared NK-EVs (5 µl) for 20 min. The grid was then washed with PBS and floated on 50 µl of 1% glutaraldehyde in PBS for 5 additional mi. The grid was washed with deionized water and incubated with 2% uranyl-oxalate on ice. Excess of reagents was removed and dried at room temperature (RT). Samples were examined using a JEOL Jem1010 operating at 100 kV, and acquired with a digital camera Gatan SC200, using the DigitalMicrograph software.

## RNA extraction, real-time PCR, small RNA NGS, and data analysis

RNA from purified human NK cells, both resting and activated in vitro (as described above), and their secreted small EVs were extracted using the miRNeasy Mini Kit (QIAGEN).

For small RNA sequencing, RNA integrity was checked using an Agilent 2100 Bioanalyzer (Agilent) for total RNA (RNA nano-chips) and for small RNA (small RNA chips), and concentrations were measured in a Nanodrop-1000 and using the Quantifluor RNA system (Promega). Samples from five healthy donors were analysed, and small RNA libraries were generated using the NEBNext Small RNA Library Prep Set from Illumina. Single read NGS was performed using an Illumina HiSeq 2500 System. Small RNASeq data were analysed by the Bioinformatics Unit at CNIC. Sequencing reads were processed as previously described (*Fernández-Messina et al., 2020*). Changes in small RNA expression were considered significant when Benjamini and Hochberg adjusted p-value < 0.05.

For qPCR analysis, isolated RNA was retro-transcribed using either the miRCURY LNA Universal RT miRNA PCR System (EXIQON) for miRNA or the Promega RT kit for mRNA, as described in *Fernández-Messina et al., 2020*. Real-time PCR was performed in a CFX384 Real-time System (Bio-Rad) using SYBR Green PCR Master Mix (Applied Biosystems). Reactions were analysed with Biogazelle QbasePlus software. Mature miRNA levels were normalized to the small nucleolar RNAs *RNU5G* and *RNU1A1*, whereas mRNA levels were normalized to glyceraldehyde-3-phosphate dehydrogenase (*GAPDH*) and *β-actin* and expressed as a relative variation from control levels. All primers for miRNA analysis were purchased from EXIQON, and primers used for mRNA detection (Metabion) are listed in supplementary methods.

## In silico target analysis

Putative mRNA targets for human miRNAs overexpressed in small EVs compared to secreting NK cells were identified using the prediction algorithm miRTarBase and are summarized in *Supplementary file 2*.

Functional unbiased analysis of EV-enriched miRNAs (compared to secreting NK cells) was performed using the IPA.

## NK-EVs functional assays

### NK-EVs and CD4⁺ T cell function assays

CD4⁺ T cells were isolated from non-adherent PBMCs, using the human CD4⁺ T cell isolation kit (MACS Miltenyi Biotec). Isolated T helper cell purity was checked by flow cytometry CD4 staining and they were then cultured in CD3 (5 µg/ml) and CD28 (2 µg/ml) coated plates either in the presence or absence of NK-EVs and both in non-polarizing (RPMI supplemented with 10% EV-free FBS) and Th1-polarizing conditions, adding a mixture of cytokines (20 U/ml IL-2 and 20 ng/ml IL-12). NK-EVs

obtained from $25 \times 10^6$ NK cells supernatants, as described above, were resuspended in 100 µl of EXO-free RPMI and 20 µl were added to $5 \times 10^6$ CD4$^+$ T cells per condition. On days 3 and 6 post-coculture, cells in the different conditions and their supernatants were analysed as described below. NK-EVs isolated from three different healthy donors were cocultured with Th cells from at least two different donors.

For EV-uptake blockade experiments, isolated CD4$^+$ T cells were supplemented with 80 µM Dynasore (Sigma-Aldrich) before NK-EV incubation.

## NK-EVs and monocyte/moDC function assays

Human moDCs were obtained as described (*Sallusto and Lanzavecchia, 1994*). Briefly, CD14$^+$ monocytes were isolated from total PBMCs, using a positive selection kit (StemCell) and cultured in RPMI medium supplemented with 10% EV-free FBS, either in the presence or absence of DC-polarizing cytokines (50 ng/ml GM-CSF and 1000 U/ml IL-4). The effects of NK-EVs addition (25 µl of isolated NK-EVS were used to treat $3 \times 10^6$ CD14$^+$ cells) were analysed 6 days after culture. NK-EVs isolated from three different healthy donors were cocultured with CD14$^+$ cells from at least two different donors.

## Flow cytometry

Cultured cells were stained with the antibodies listed in *Supplementary file 3* and analysed by flow cytometry in a BD FACSCanto cytometer. For all analysis, only live single cells were included and analysed with the FlowJov10 software. Dead cells were excluded by DAPI or Live/Dead Fixable Yellow (Thermo Fisher) staining. Murine cells were incubated with mouse CD16/CD32/Fc shield and human cells with 100 µg/ml γ-globulin before antibody staining.

## Immunoblotting

For Western blot analyses, cell lysates were prepared by incubation in TNE buffer containing 1% NP-40 and a protease inhibitor cocktail (Complete, Roche). Nuclei were eliminated by centrifugation at 13,000× *g*. Lysates and membrane-soluble fractionations were run on 10% SDS-PAGE gels and transferred to Immobilon-P (Millipore) membrane. The membrane was blocked using PBS containing 0.1% Tween-20 (PBS-T) and 5% non-fat dry milk. Detection of proteins was performed by incubation with the appropriate specific primary and secondary antibodies (*Supplementary file 3*). Proteins were visualized using the ECL system (Amersham Pharmacia) and chemoluminiscence was measured with LAS-3000 (Fujifilm).

For dot blot analyses, 2 µl of isolated NK EVs were blotted onto nitrocellulose membranes (Amersham) and proteins were visualized as described for Western blot analyses. Protein concentration was quantified using the ImageJ software.

## Cytokine arrays

CD4$^+$ T cells were isolated as described above from human buffy coats. A total of $5 \times 10^6$ cells were incubated for 16 hr with NK-EVs isolated from $5 \times 10^6$ 72 hr NK cell cultures in RPMI supplemented with 10% EV-free FBS. Culture supernatants were analysed using the Proteome Profiler Human Cytokine Array Kit (R&D Systems), following the manufacturer's instructions.

## Cytokine analysis of cell culture supernatants by ELISA

Cytokine production was detected in cell culture supernatants using ELISA kits purchased from DIACLONE (IL-2) and eBioscience (IFN-γ), respectively, following the manufacturer's instructions. Absorbance was measured at 450 nm, with a reference wavelength of 570 nm.

## Nanoparticle-miRNAs synthesis

AuNPs were synthesized following the Turkevich method (*Turkevich, 1951*). Rounded 12 nm AuNPs at 25.2 nM were obtained and modified with oligonucleotides (IDT) containing the proper thiol modifications (*Supplementary file 4*). MiRNA duplex formation was performed using the same volume and concentration of both strands diluted in annealing buffer (10 mM Tris, pH 7.5–8.0, 50 mM NaCl, 1 mM EDTA), according to Sigma Aldrich's protocol (https://www.sigmaaldrich.com/ES/es/technical-documents/protocol/genomics/pcr/annealing-oligos). The mixture was incubated for 10 min at 95°C and

then cooled down to RT slowly. Before conjugation, the miRNA duplexes were deprotected by incubation with ×100 excess TCEP for 2 hr at RT. For the functionalization reaction, 4 pmol/ µl AuNPs of deprotected miRNA duplex were added to 1 ml AuNP for each condition. AuNP miR-mix contains 4 pmol/µl of each miRNA duplex; AuNP control contains 4 pmol/µl poli T-thiol (TTTTTTTTTTT/3ThioMC3-D/). After oligonucleotide addition, NaCl was added in small volumes until 0.3 M final concentration. Finally, the AuNPs were incubated 16 hr in agitation, light protected at RT. After this step, three washes were performed by 40 min 16,100× *g* centrifugation at 4°C. In the last washing step, AuNPs were resuspended in sterile ×1 PBS and filtered with 0.2 µm syringe-filter in a laminar hood.

## Nanoparticle-miRNAs analysis

Nanoparticles in PBS were injected into the footpad (30 µl) of wild-type C57BL/6 mice. On day 5, animals were sacrificed and splenocytes isolated and analysed by flow cytometry and qRT-PCR before and after stimulation with anti-CD3 and anti-CD28 antibodies. All cells were incubated with PMA (50 ng/ml) and ionomycin (500 ng/ml) 4 hr before analysis, and Brefeldin A (5 µg/ml) was added to block secretion in flow cytometry experiments. Supernatants from cultured cells after 16 hr culture were also analysed by ELISA.

## Small RNASeq analyses

Small RNASeq data were analysed by the Bioinformatics Unit at CNIC. Sequencing reads were pre-processed by means of a pipeline that used FastQC (http://www.bioinformatics.babraham.ac.uk/projects/fastqc/), to asses read quality, and Cutadapt to trim sequencing reads, eliminating Illumina adaptor remains, and to discard those that were shorter than 20 nucleotides after trimming. Around 60% of the reads from any of the samples were retained. Resulting reads were aligned against a collection of 2657 human, mature miRNA sequences extracted from miRBase (release 21), to obtain expression estimates with RSEM (*Gutiérrez-Vázquez et al., 2017*). Percentages of reads participating in at least one reported alignment were around 22%. Expected expression counts were then processed with an analysis pipeline that used Bioconductor package Limma (*Koyano et al., 2021*) for normalization (using TMM method) and differential expression testing, considering only 708 miRNA species for which expression was at least 1 count per million (CPM) in three samples. Changes in gene expression were considered significant if the Benjamini-Hochberg adjusted p-value < 0.05.

## miRNA PtMs analyses

Epi-transcriptomic modifications were detected with Chimira (*Koppers-Lalic et al., 2014*), an online tool that, after alignment of miRNASeq reads against miRBAse21 records, identifies mismatched positions to classify them and to quantify multiple types of 3'-modifications (e.g. uridylation), as well as internal and 5'-modifications. Count tables produced by Chimira were further processed with ad hoc produced R-scripts to calculate summary statistics across groups of replicate samples.

## Motif enrichment analysis

Over-represented motifs in miRNA collections were detected with MEME (*Bailey et al., 2015*), using the differential enrichment method and complementary miRBAse21 miRNA collections as background references. Both types of input were previously processed with USEARCH to make them non-redundant at 80% identity (*Edgar, 2010*). Motif searches were performed only in the forward strand, using a motif size of 4–8 nt, zero-order Markov model for nucleotide distributions, and expecting zero or one motif occurrence per sequence (ZOOPS).

## Statistical analyses

All data were analysed with GraphPad Prism 5.0 and 8.0 (GraphPad Software, San Diego, CA). All data were included in the analysis, unless identified as outliers, using GraphPad algorithms. Appropriate statistical tests were used, for each experiment, as indicated. Significance was set at *p<0.05, **p<0.01, ***p<0.001.

# Acknowledgements

NGS experiments were performed in the CNIC Genomics Unit (Centro Nacional de Investigaciones Cardiovasculares, Madrid, Spain) and analysed by the CNIC Bioinformatics Unit. This manuscript

was funded by grants PDI-2020-120412RB-I00 and PDC2021- 121719-I00 (FS-M) and PID2020-119352RB-I00 (AS) from the Spanish Ministry of Economy and Competitiveness; CAM (S2017/BMD-3671-INFLAMUNE-CM) from the Comunidad de Madrid (FS-M). CIBERCV (CB16/11/00272) and BIOIMID PIE13/041 from the Instituto de Salud Carlos. The current research has received funding from 'la Caixa' Foundation under the project code HR17-00016. Grants from Ramón Areces Foundation 'Ciencias de la Vida y de la Salud' (XIX Concurso-2018) and from Ayuda Fundación BBVA y Equipo de Investigación Científica (BIOMEDICINA-2018) (to FSM). The CNIC is supported by the Ministerio de Ciencia, Innovacion y Universidades and the Pro-CNIC Foundation, and is a Severo Ochoa Center of Excellence (SEV-2015–0505). IMDEA Nanociencia acknowledges support from the 'Severo Ochoa' Programme for Centres of Excellence in R&D (MINECO, CEX2020-001039-S). SGD is supported by a grant from the Spanish Ministry of Universities. Authors thank Dr Miguel Vicente-Manzanares for critical review and editing. We also thank Dr Francisco Urbano and Dr Covadonga Aguado for their support with EM (TEM facilities, Universidad Autónoma de Madrid).

## Additional information

### Funding

| Funder | Grant reference number | Author |
| --- | --- | --- |
| Spanish National Plan for Scientific and Technical Research and Innovation | PD1-2020-120412RB-100 | Francisco Sánchez Madrid |
| Fundación BBVA | | Francisco Sánchez Madrid |
| Comunidad de Madrid | | Francisco Sánchez Madrid |
| Ministry of Economy | | Francisco Sánchez Madrid |

The funders had no role in study design, data collection and interpretation, or the decision to submit the work for publication.

### Author contributions

Sara G Dosil, Conceptualization, Data curation, Formal analysis, Investigation, Methodology, Writing – original draft; Sheila Lopez-Cobo, Milagros Castellanos, Conceptualization, Investigation, Methodology, Writing – review and editing; Ana Rodriguez-Galan, Data curation, Formal analysis, Investigation, Writing – review and editing; Irene Fernandez-Delgado, Investigation, Methodology, Writing – review and editing; Marta Ramirez-Huesca, Paula Milan-Rois, Investigation, Methodology; Alvaro Somoza, Conceptualization, Funding acquisition, Methodology, Writing – review and editing; Manuel José Gómez, Data curation, Software, Formal analysis, Methodology; Hugh T Reyburn, Mar Vales-Gomez, Conceptualization, Methodology, Writing – review and editing; Francisco Sánchez Madrid, Conceptualization, Supervision, Funding acquisition, Writing – original draft, Project administration, Writing – review and editing; Lola Fernandez-Messina, Conceptualization, Formal analysis, Validation, Investigation, Methodology, Writing – original draft, Writing – review and editing

### Author ORCIDs
Ana Rodriguez-Galan ⓘ http://orcid.org/0000-0001-6209-782X
Paula Milan-Rois ⓘ http://orcid.org/0000-0002-7043-2920
Francisco Sánchez Madrid ⓘ http://orcid.org/0000-0001-5303-0762
Lola Fernandez-Messina ⓘ http://orcid.org/0000-0002-2163-8746

### Ethics
All experimental methods and protocols were approved by the CNIC and the Comunidad Autónoma de Madrid and conformed to European Commission guidelines and regulations (PROEX-206.1/20).

### Decision letter and Author response
Decision letter https://doi.org/10.7554/eLife.76319.sa1
Author response https://doi.org/10.7554/eLife.76319.sa2

## Additional files

### Supplementary files

• Supplementary file 1. Small RNA sequencing of resting natural killer (NK) and activated NK cells and their derived NK-extracellular vesicles (EVs). Table summarizes the analysis of samples generated from five donors, showing differentially expressed microRNAs (miRNAs). Provided as an additional Excel file.

• Supplementary file 2. Natural killer-derived extracellular vesicle (NK-EV) microRNAs (miRNAs) mRNA target candidates. Table summarizes putative mRNA targets for selected NK-EV miRNAs identified by in silico analyses, using the miRTarBase database. T cell function-related miRNAs are highlighted in orange and dendritic cell (DC)-related miRNAs in blue.

• Supplementary file 3. Antibodies and primers. Specific primers for the detection of mRNA related with T cell functions are shown in orange and dendritic cell (DC) function-related miRNAs in blue. Housekeeping mRNAs are shown in grey.

• Supplementary file 4. Oligonucleotides for microRNAs (miRNA) duplex gold nanoparticles (AuNPs) design.

• Transparent reporting form

### Data availability

Sequencing data have been deposited in the Gene Expression Omnibus and are available to readers under record GSE185171. EV isolation procedures are available at EV-TRACK knowledgebase (EV-TRACK ID: EV210234).

The following dataset was generated:

| Author(s) | Year | Dataset title | Dataset URL | Database and Identifier |
|---|---|---|---|---|
| Sánchez Madrid F | 2022 | Natural killer (NK) cell-derived extracellular-vesicle shuttled microRNAs control T cell responses | http://www.ncbi.nlm.nih.gov/geo/query/acc.cgi?acc=GSE185171 | NCBI Gene Expression Omnibus, GSE185171 |

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
