## [Editor Report]

This report identified NK-extracellular-vesicle (NK-EV)-associated microRNAs and characterized them by small RNA next-generation sequencing. They found that NK-EVs promote Th1 polarization and activation of monocyte and moDCs. The findings are potentially important for understanding NK cell function.

---

## [Decision Letter]

**Decision letter after peer review:**

Thank you for submitting your article "Natural killer (NK) cell-derived extracellular-vesicle shuttled microRNAs control T cell responses" for consideration by *eLife*. Your article has been reviewed by 3 peer reviewers, one of whom is a member of our Board of Reviewing Editors, and the evaluation has been overseen by Tadatsugu Taniguchi as the Senior Editor. The following individual involved in the review of your submission has agreed to reveal their identity: Michiel Pegtel (Reviewer #2).

Essential revisions:

1) Because the authors have not performed size-exclusion or density gradient separation before applying their functional experiments with purified EVs, they cannot rule out that other molecules in the activated NK secretome are responsible for the observed effects. For example, NK-EVs contain several effector proteins including IFN-γ (Choi et al., Molecules, 2020;25(21)). The authors should therefore exclude the possibility of contamination of IFN-γ derived from NK-EVs in Figure 4. Blocking EV uptake with dynasore for example could rule out a major contribution from other soluble factors. Since differential ultracentrifugation only is prone to have contamination of protein complexes, lipoparticles etc, can the authors show EM data of their EVs? In addition, how do the authors exclude the presence of feeder cell EVs in their EV-preps?

2) The suppl data shows some Western blotting of EVs from activated NK cells. Apart from their DE miRNA analysis between EVs from resting vs activated NK cells, it is possible that resting NK cells secrete much less EVs, it would behove the authors to demonstrate this with Western and NTA or TPRS.

The authors validated selected miRNAs expression by qPCR in Figure 1E. However, no miRNAs were significantly enriched in NK-EVs compared with activated NK cells. To support the author's proposal, the statistical significance of their expression between activated NK cells and NK-EVs should be confirmed.

Maybe the authors can include a separate table of the miRNAs that change in the activated NKs cells? In addition, log fold changes (alone) can be deceiving. Maybe the authors can show (relative) abundance data of their EV-enriched miRNAs?

3) Because the authors have not used control EVs from other cell types it remains somewhat unclear whether all EVs or specifically EVs from (activated) NK cells have the polarizing effects on the recipient cells. Furthermore, the authors proposed that NK-EV-associated miRNAs promote Th1-like responses via T-bet de-repression by downregulating Gata3 mRNA. To support their proposal, double-staining of T-bet and GATA3 by flow cytometry should be conducted in Figure 4 and Figure 7. In Figure 8, the authors claimed that NK-EV-associated miRNAs promoted Th1-like responses. However, it seems to be an overestimation because they did not show the downregulation of GATA3 by NK-EV-associated miRNAs and T-bet de-repression as well as IFN-γ induction from CD4^+^ T cells by treatment with NK-EV-associated miRNAs. This fails to support the story of this manuscript.

Perhaps the authors could use an irrelevant control cell type that does not express much less of their miRNAs. As the nanoparticle experiments strongly suggest the involvement of miRNAs (in vivo), the authors should use their nanogold particles with miRNAs as controls for some of the crucial in vitro NK-EVs experiments discussed above and to determine what candidate miRNAs have the effect.

4) Template-independent terminal ribonucleotide transferases (TENTs) catalyze the addition of nucleotide monophosphates to the 3′-end of RNA molecules. While their efforts in their pMT (isomiR may be better) analysis are appreciated, global analysis of all miRNAs are not very informative. It is surprising that so many cytosine additions are found whereas adenylation and uridylation are known to be caused by uridylases and adenylases. Is there evidence of 3' tagging of mature mammalian miRNAs by cytidylation. Obviously, it would be interesting if this type of modification is somehow NK cell function related.

5) Despite a large body of independent evidence, the biological relevance of many Isomirs as found in sequencing data has recently been scrutinized (Hu et al., Nature Biotech 2021). Perhaps the authors can comment in their paper why their results are unlikely the result of artifacts during library preparation?

6) Their analysis of the NTAs, though comprehensive, makes little sense to me. The authors state that they analyzed variations from position -4 to +4. If the average mature miRNA is 21nts long, they thus examined sequences of 17-25 nucleotides, yet in their methods, they stated that they 'filter' the reads from 20nts? Please clarify.

How is it possible that position 0 (the last nucleotide on the 3' end of the miRNA) has a nucleotide addition if anything it should be a substitution? The authors should show the percentage of fully mature (untagged) canonical miRNAs in their graphs. The first NTA should be position +1, etc.

7) Figure 3B is interesting in that some miRNAs such miR122-5p is for 50% modified in resting cells yet not at all in activated cells, yet the (activated) NK cell EVs do contain modified miRNAs. What are the absolute or relative levels of these miRNAs in the NK cells? Can the authors provide an explanation?

The authors demonstrated that NK-EV-associated miRNAs possessed unique post-transcriptional modifications and short motifs in Figure 2 and Figure 3. Since the conclusions from both figures are similar, it would be better to combine Figure 2 and Figure 3 together.

*Reviewer #1 (Recommendations for the authors):*

Because it is unclear that the features of NK-EVs described in this paper is specific to NK-EVs, I recommend that the authors include data showing how NK cell-derived EVs are characterized compared to EVs derived from other cells. The followings are specific points that may improve the paper.

1. Are there differences in the repertoire of miRNAs among different subsets of lymphocytes?

2. Is the promotion of Th1-like responses specific to NK-EVs?

3. Similarly, is the down-modulation of Gata-3 and T-bet specific to NK EVs?

4. EVs are produced by many cell types. Is it possible to estimate the proportion of NK-EVs in total EVs?

*Reviewer #2 (Recommendations for the authors):*

1) The authors isolated EVs with differential ultracentrifugation only which is prone to contamination of protein complexes, lipoparticles etc. Can the authors show EM data of their EVs.

2) Because the authors have not performed size-exclusion nor density gradient separation before applying their functional experiments with purified EVs they cannot rule out that other molecules in the activated NK secretome are responsible for the observed effects? Blocking EV uptake with dynasore for example could rule out a major contribution from other soluble factors.

3) The suppl data shows some western blotting of EVs from activated NK cells. Apart from their DE miRNA analysis between EVs from resting vs activated NK cells, it is possible that resting NK cells secrete much less EVs, it would behove the authors to demonstrate this with western and NTA or TPRS.

4) How do the authors exclude the presence of feeder cell EVs in their EV-preps?

5) Because the authors have not used control EVs from other cell types it remains somewhat unclear whether all EVs or specifically EVs from (activated) NK cells have the polarizing effects on the recipient cells. Perhaps the authors could use an irrelevant control cell type that does not express much less of their miRNAs. I do appreciate that the nanoparticle experiments strongly suggest the involvement of miRNAs (in vivo). The authors should use their nanogold particles with miRNAs as controls for some of the crucial the in vitro NK-EVs experiments. Do their nanoparticles have similar effects on CD4^+^ Th1 polarization and what candidate miRNAs have this effect? Ideally, the authors would examine the effects of NK-EVs in mice but I find that not required.

6) Maybe the authors can include a separate table of the miRNAs that change in the activated NKs cells?

7) Log fold changes (alone) can be deceiving. Maybe the authors can show (relative) abundance data of their EV-enriched miRNAs?

8) Template-independent terminal ribonucleotide transferases (TENTs) catalyze the addition of nucleotide monophosphates to the 3′-end of RNA molecules. While I appreciate their efforts in their pMT (I prefer isomiR) analysis, global analysis of all miRNAs are in my view not very informative. It is surprising that so many cytosine additions are found whereas adenylation and uridylation are known to be caused by uridylases and adenylases. Is there evidence of 3' tagging of mature mammalian miRNAs by cytidylation. Obviously, it would be interesting if this type of modification is somehow NK cell function related.

9) Despite a large body of independent evidence, the biological relevance of many Isomirs as found in sequencing data has recently been scrutinized (Hu et al., Nature Biotech 2021). Perhaps the authors can comment in their paper why their results are unlikely the result of artifacts during library preparation?

10) Their analysis of the NTAs, though comprehensive, makes little sense to me. The authors state that they analyzed variations from position -4 to +4. If the average mature miRNA is 21nts long, they thus examined sequences of 17-25 nucleotides, yet in their methods, they stated that they 'filter' the reads from 20nts? Please clarify.

11) How is it possible that position 0 (the last nucleotide on the 3' end of the miRNA) has a nucleotide addition if anything it should be a substitution? The authors should show the percentage of fully mature (untagged) canonical miRNAs in their graphs. The first NTA should be position +1, etc.

12) Figure 3B is interesting in that some miRNAs such miR122-5p is for 50% modified in resting cells yet not at all in activated cells, yet the (activated) NK cell EVs do contain modified miRNAs. What are the absolute or relative levels of these miRNAs in the NK cells? Can the authors provide an explanation?

*Reviewer #3 (Recommendations for the authors):*

In the report entitled "Natural killer (NK) cell-derived extracellular-vesicle shuttled microRNAs control T cell responses", Dosil et al., identified NK-extracellular-vesicle (EV)-associated microRNAs and their post-transcriptional modifications signature by small RNA next-generation sequencing. They also found that NK-EVs promoted Th1 polarization and activation of monocyte and moDCs. They suggested that the identified NK-EV-associated microRNAs partially recapitulated NK-EV effects in T cells in vivo.

The study contains some interesting findings. However, the impacts of NK-EVs and NK-EV-associated microRNAs on Th1 differentiation are not impressive. In addition, their evidence is not sufficient to support their proposal that NK-EV-associated microRNAs promote Th1-like responses via T-bet de-repression by down-regulation of GATA3.

1. The authors validated selected microRNAs expression by qPCR in Figure 1E. However, no microRNAs were significantly enriched in NK-EVs compared with activated NK cells. To support the author's proposal, statistical significance of their expression between activated NK cells and NK-EVs should be confirmed.

2. The authors demonstrated that NK-EV-associated microRNAs possessed unique post-transcriptional modifications and short motifs in Figure 2 and Figure 3. Since the conclusions from both figures are similar, it would be better to combine Figure 2 and Figure 3 together.

3. As mentioned above, the impact of NK-EVs on Th1 polarization was quite minor in Figure 4. Furthermore, the authors failed to induce IFN-γ production from CD4^+^ T cells by administration of selected microRNAs in Figure 7F.

NK-EVs contain several effector proteins including IFN-γ (Choi et al., Molecules, 2020;25(21)). So, they should exclude the possibility of contamination of IFN-γ derived from NK-EVs in Figure 4.

4. The authors proposed that NK-EV-associated microRNAs promote Th1-like responses via T-bet de-repression by downregulating Gata3 mRNA. To ensure their proposal, double-staining of T-bet and GATA3 by flow cytometry should be conducted in Figure 4 and Figure 7.

5. In Figure 8, the authors represent that NK-EV-associated microRNAs promote Th1-like responses. However, it seems to be an overestimation because they did not show the downregulation of GATA3 by NK-EV-associated microRNAs and T-bet de-repression as well as IFN-γ induction from CD4^+^ T cells by treatment of NK-EV-associated microRNAs. This fails to support the story of this manuscript.

---

## [Author Response]

Essential revisions:1) Because the authors have not performed size-exclusion or density gradient separation before applying their functional experiments with purified EVs, they cannot rule out that other molecules in the activated NK secretome are responsible for the observed effects. For example, NK-EVs contain several effector proteins including IFN-γ (Choi et al., Molecules, 2020;25(21)). The authors should therefore exclude the possibility of contamination of IFN-γ derived from NK-EVs in Figure 4. Blocking EV uptake with dynasore for example could rule out a major contribution from other soluble factors. Since differential ultracentrifugation only is prone to have contamination of protein complexes, lipoparticles etc, can the authors show EM data of their EVs? In addition, how do the authors exclude the presence of feeder cell EVs in their EV-preps?

We agree with the reviewers that ultracentrifugation of cell culture supernatants does not allow to fully rule out the involvement of other molecules, such as IFN-γ in the observed effects. To address this issue we have performed additional experiments, using different approaches, as suggested by the reviewers.

First, we analyzed the effects of NK-derived EVs on isolated T lymphocytes pre-treated with Dynasore to block EV uptake. As included in the new figure (new Figure 3—figure supplement 2A,B and Figure 4—figure supplement 2A,B), after dynamin-dependent endocytosis blockade, the upregulation of IFN-γ and CD25 and driven by NK-EVs were abolished, indicating that the observed effects on T cell function are mediated by NK-derived EVs. It is worthwhile mentioning, that since dynasore has been shown to enhance the formation of mitochondrial antiviral signalling aggregates and promote endocytosis-independent NF-κB activation, this may interfere with lymphocyte activation (1).

To further rule out the involvement of soluble factors co-precipitating with the small EV pellet after 100,000 g ultracentifugation, we isolated small EVs from NK cell culture supernatants, using size-exclusion chromatography (SEC) (new Figure 3—figure supplement 2C-H and new Figure 4figure supplement 2C,D). Fractions 8-11, that expressed the exosomal markers CD81 and Tsg101 and low levels of protein aggregates (new Figure 3—figure supplement 2C,D), were pooled and ultracentrifuged before functional analysis. Despite a significant loss of NK-EV material during the SEC purification steps (new Figure 3—figure supplement 2E,F), confirmed by a reduction of the exosomal markers CD81 and Tsg101 to around 20% of the levels obtained by ultracentrifugation, a similar impact for SEC-isolated NK-derived small EVs on T cell function was observed, both in IFN-γ (Figure 3—figure supplement 2G,H) and more significantly in CD25 expression (Figure 4—figure supplement 2C,D).

Altogether, experiments using either Dynasore or SEC, confirmed that NK-EVs account for the observed effects, although we cannot completely rule out that the presence of soluble factors may also play a partial role. These experiments have been added to the manuscript as supplementary materials, as indicated, and the main text, discussion and material methods have been modified accordingly, as highlighted in the revised version of the manuscript.

Additionally, as suggested by the reviewers, we have performed Electron microscopy from NKEVs isolated from NK cell culture supernatants by differential ultracentrifugation, as described in Figure 1—figure supplement 1A and methods. Electron microscopy confirmed the average size range of small EVs (between 50-200nm) and their typical cup-shape morphology and membranous structure by negative staining (Figure 1—figure supplement 1H). The main text of the manuscript and the material and methods section have been modified accordingly.

Concerning the reviewer´s comment, we acknowledge that the presence of feeder cell-derived EVs in the vesicle preparation needs to be avoided. For this reason, our protocol rules out this possibility by isolating NK cells expanded in the presence of the cytokine cocktail and feeder cells, as described in methods, before small EV accumulation and isolation. After NK cell isolation, using the appropriate Miltenyi kit, following the manufacturer´s indications, the purity of isolated NK cells was checked by flow cytometry (CD3^-^ CD56^+^), before incubating the cells in EV-depleted medium for small EV accumulation and further isolation by ultracentrifugation. This has been further clarified in the main text of the revised manuscript.

2) The suppl data shows some Western blotting of EVs from activated NK cells. Apart from their DE miRNA analysis between EVs from resting vs activated NK cells, it is possible that resting NK cells secrete much less EVs, it would behove the authors to demonstrate this with Western and NTA or TPRS.The authors validated selected miRNAs expression by qPCR in Figure 1E. However, no miRNAs were significantly enriched in NK-EVs compared with activated NK cells. To support the author's proposal, the statistical significance of their expression between activated NK cells and NK-EVs should be confirmed.Maybe the authors can include a separate table of the miRNAs that change in the activated NKs cells? In addition, log fold changes (alone) can be deceiving. Maybe the authors can show (relative) abundance data of their EV-enriched miRNAs?

Resting NK cells were isolated form healthy donor´s buffy coats and, in parallel, activated NK cells were generated using our standard procedure, as described in the methods section. Small EVs were isolated from both resting and activated NK cells by ultracentrifugation, as described in the manuscript. It is important to highlight that resting CD3^-^CD56^+^ NK cells are a small proportion of peripheral-blood mononuclear cells, representing according to Angelo et al., 0.61 – 16.87% in healthy adults, with a mean of 6.47% (2). Thus, the number of resting NK cells that we could isolate from buffy coats were very limited (up to 5x10^6^ total cells) and therefore the amounts of small EVs that we could isolate were very low, even from activated NK cells, and unfortunately below the limit of detection of our western blotting techniques.

Nevertheless, using Nanoparticle-tracking analysis (NTA), we could detect small EVs derived from 5x10^6^ either resting or in vitro activated NK cells, although at very low levels, and confirmed that activated NK cells release significantly more small EVs than their resting cell counterparts (Figure 1—figure supplement 1I). Indeed, very few particles were detected in resting NK cell-derived small EVs. Moreover, dot blot analyses allowed to detect CD81 and Tsg101 exosomal markers in small EVs purified from activated NK cells, while EVs released by the same number of resting NK cells could not be detected (Figure 1—figure supplement 1J). Altogether our data confirm that resting NK cells release much less EVs than activated NK cells.

Following the reviewers comments we have included a separate table where we show the relative abundance of EV-miRNAs in secreting activated NK cells and their secreted EVs from small RNA sequencing data, and the corresponding plots, including statistics (new Figure 1—figure supplement 2B,C). Also, we have re-analyzed the qPCR data and show Paired t-test analysis, comparing Resting vs Activated cells, Activated cells vs NK-EVs and Resting cells vs NK-EVs (Figure 1E,F). These analyses confirm the results from the sequencing data, showing differences between Activated NK cells and their released NK-EVs, that reach statistic significance in most cases. For hsa-miR-10b-5p and hsa-miR-99a-5p validations by qPCR, comparing activated cells and their released EVs, t-tests yields p-values slightly above 0.1 (0.1656 and 0.1302, respectively), due to the variability among donors. However, the enrichment of these miRNAs in NK-EVs compared to their parental cells is consistent within individuals, as shown in Figure 1—figure supplement 2 and Supplementary Table S1.

3) Because the authors have not used control EVs from other cell types it remains somewhat unclear whether all EVs or specifically EVs from (activated) NK cells have the polarizing effects on the recipient cells. Furthermore, the authors proposed that NK-EV-associated miRNAs promote Th1-like responses via T-bet de-repression by downregulating Gata3 mRNA. To support their proposal, double-staining of T-bet and GATA3 by flow cytometry should be conducted in Figure 4 and Figure 7. In Figure 8, the authors claimed that NK-EV-associated miRNAs promoted Th1-like responses. However, it seems to be an overestimation because they did not show the downregulation of GATA3 by NK-EV-associated miRNAs and T-bet de-repression as well as IFN-γ induction from CD4^+^ T cells by treatment with NK-EV-associated miRNAs. This fails to support the story of this manuscript.Perhaps the authors could use an irrelevant control cell type that does not express much less of their miRNAs. As the nanoparticle experiments strongly suggest the involvement of miRNAs (in vivo), the authors should use their nanogold particles with miRNAs as controls for some of the crucial in vitro NK-EVs experiments discussed above and to determine what candidate miRNAs have the effect.

To address this key point raised by the reviewers, several new experiments were performed.

First, small EVs from two distinct human cell lines (namely the HEK-293, human epithelial kidney cells and the Raji B lymphoblast cells) were isolated, following the differential ultracentrifugation protocol, as described in the methods section. Their effects in primary T cells isolated from human healthy donors showed no impact, neither in IFN-γ secretion (new Figure 3—figure supplement 3), nor in activation, measured by CD25 expression (Figure 4—figure supplement 2E,F), that even decreased upon Raji B cell EV-treatment under Th1-polarizing conditions.

Also, three microRNAs that are preferentially excluded from the NK-EV fraction were selected, namely hsa-miR-124, hsa-miR-3667 and hsa-miR-4158 and loaded onto gold-nanoparticles (new Figure 6—figure supplement 2), and their effects were evaluated in immunocompetent C57/BL6 mice after footpad injection. These experiments showed no effects of these nanoparticles, as observed for NK-EV enriched microRNAs, neither in activation, nor in IFN-γ secretion (new Figure 6H).

Furthermore, we have analyzed the effects of the addition of a mixture of hsa-miR-92a-3p and hsa-miR-10b-5p AuNPs to CD4^+^ T cells, that putatively target GATA-3 (Supplementary Table 2). Three days after incubation, we found that, as observed for mRNA transcript levels (Figure 3E), GATA-3 protein levels were reduced while T-bet expression increased, compared to CD4^+^ T cells incubated with Control-AuNPs by western blot (Figure 3—figure supplement 1F) and a similar trend was observed by flow cytometry analyses after NK-EV incubation (Figure 3—figure supplement 1E).However, since it is well documented that miRNAs target several mRNAs, very likely other targets may contribute to the observed effects.

4) Template-independent terminal ribonucleotide transferases (TENTs) catalyze the addition of nucleotide monophosphates to the 3′-end of RNA molecules. While their efforts in their pMT (isomiR may be better) analysis are appreciated, global analysis of all miRNAs are not very informative. It is surprising that so many cytosine additions are found whereas adenylation and uridylation are known to be caused by uridylases and adenylases. Is there evidence of 3' tagging of mature mammalian miRNAs by cytidylation. Obviously, it would be interesting if this type of modification is somehow NK cell function related.

Following the reviewer´s suggestion we have modified in the text to refer to modified miRNAs as isomiRs. As the reviewer points out, uridylation and adenylation are the most typical 3′ end modifications across animal miRNAs (3-6).

Several studies evaluating PtMs have found guanidinylation and cytosilation to be barely represented. However, cytosine addition was found to be the second most abundant 3´modification in mouse primordial germ cells and gonadal somatic cells at various embryonic stages (7) and was also described in Arabidopsis (8).

Moreover, in a recent paper from our group, cytosylation was found to be a significant posttranscriptional modification in human T cells (9), suggesting that cytosine additions may be relevant for human lymphocyte function.

5) Despite a large body of independent evidence, the biological relevance of many Isomirs as found in sequencing data has recently been scrutinized (Hu et al., Nature Biotech 2021). Perhaps the authors can comment in their paper why their results are unlikely the result of artifacts during library preparation?

We acknowledge that ligation bias has been well documented in conventional small RNA-seq methods. It has thus been suggested that non-canonical “modification tails” may be an artifact due to the specific 3’ adaptor sequence and/or ligation chemistry used for library preparation.

In our case, the adaptor remains cannot contribute to the presence of non-templated "C" residues at the 3' end of miRNAs, as described in Rodríguez-Galan et al., (9). Briefly, in our experimental settings, adaptors used in the current experiments are standard Illumina TruSeq adaptors that start with the sequence "AGATCGGAAGAGCACACGTCT". Adaptor sequences were removed with cutadapt, using the following line of code, for each fastq file:

cutadapt -m 15-M35-O7-o trimmed.fq --untrimmed-output=untrimmed.fq --too-shortoutput=tooshort.fq -a AGATCGGAAGAGCACACGTCTGAACTCCAGTCAC input.fq

Regarding the possibility of ligation bias, we are aware that this may affect any library preparation strategy. While it is true that some bias might exist in the absolute quantification of miRNAs with a specific nucleotide composition, in our manuscript, we focus on comparing this composition across different types of samples (resting NK cells, activated NK cells and NK-derived EVs). Thus, this bias would be affecting every sample and our comparisons would still be valid.

6) Their analysis of the NTAs, though comprehensive, makes little sense to me. The authors state that they analyzed variations from position -4 to +4. If the average mature miRNA is 21nts long, they thus examined sequences of 17-25 nucleotides, yet in their methods, they stated that they 'filter' the reads from 20nts? Please clarify.How is it possible that position 0 (the last nucleotide on the 3' end of the miRNA) has a nucleotide addition if anything it should be a substitution? The authors should show the percentage of fully mature (untagged) canonical miRNAs in their graphs. The first NTA should be position +1, etc.

We agree with the reviewer that our analysis is based on non-templated modifications, being either substitutions within the canonical sequence (mainly positions -1, 0), or additions, from position +1 onwards. For the sake of clarity, we have modified the nomenclature in the main text and refer to each specific modification as either nucleotide “additions” or “substitutions”, as suggested by the reviewer.

The average size of mature miRNAs is 22 nt and more than 90% of those described in miRBAse21 have lengths equal to 20 nt and/or higher. For the current study, read pre-processing was performed with a pipeline that used cutadapt and FastQC: the first to remove Illumina adapters and to select trimmed reads with lengths equal or larger than 20, and the second to perform quality checks on the reads.

To identify PtMs, we used Chimira to align pre-processed reads against miRNA hairpin sequences from the human component of miRBase, with the aim of identifying nucleotide sequences diverging from the reference sequence (modifications).

To provide a common, unified framework for visualization and PtM summarization, Chimira adjusts the alignments on the 3'end, mapping everything to a length of 23 nt, using a zero-based coordinate system.

In that context, the initial position (5'end) is referred as "zero", and modification positions located before or after the 5'end are identified with negative and positive indices, respectively. The end position (3'end) is numbered as position "22", independently of the real length of miRNAs. As it happened on the 5'ends, modification positions located before or after the 3'end are identified with negative and positive indices, respectively.

Therefore, by focusing on positions -4 to +4, we were searching for sequence variations, relative to each reference sequence in miRBAse, in a 9 nt long region that was centered on the 3'end of miRNAs, independently of their length.

7) Figure 3B is interesting in that some miRNAs such miR122-5p is for 50% modified in resting cells yet not at all in activated cells, yet the (activated) NK cell EVs do contain modified miRNAs. What are the absolute or relative levels of these miRNAs in the NK cells? Can the authors provide an explanation?

We agree with the reviewer that it is interesting that certain miRNAs, namely miR-122-5p, but also miR-409-3p and miR-451a that follow a similar trend, are highly modified in resting NK cells, and in activated NK-derived EVs, while absent y activated NK cells. We believe, that a possible explanation for this observation is, that upon activation, modified miRNAs, e.g. miR-122-5p are more prone to be sorted into EVs. In this sense, it was described that uridylation may promote exosome loading, while adenylated miRNA isoforms were relatively enriched in B cells (10). Thus, a similar mechanism might be occurring in human activated NK cells for modified miRNAs, that could be tagged for EV packaging. This has been added to the Discussion section of the revised manuscript. Concerning the relative values of these miRNAs, for miR-122-5p, these levels are relatively low but very similar in both resting and activated NK cells, while high in NK-EVs (Relative Counts Per Million; REST: 0,57, ACT: 0,52 and NK-EVs: 313,94).

The authors demonstrated that NK-EV-associated miRNAs possessed unique post-transcriptional modifications and short motifs in Figure 2 and Figure 3. Since the conclusions from both figures are similar, it would be better to combine Figure 2 and Figure 3 together.

Figures 2 and 3 have been combined in New Figure 2 as suggested by the reviewers and supplementary panels organized in the corresponding figure supplements.

Reviewer #1 (Recommendations for the authors):Because it is unclear that the features of NK-EVs described in this paper is specific to NK-EVs, I recommend that the authors include data showing how NK cell-derived EVs are characterized compared to EVs derived from other cells. The followings are specific points that may improve the paper.1. Are there differences in the repertoire of miRNAs among different subsets of lymphocytes?

Please refer to the revision reply to reviewer points 1 and 2.

2. Is the promotion of Th1-like responses specific to NK-EVs?

Please refer to the revision reply to reviewer points 1 and 2. Indeed, EVs derived from other cell types, including the lymphocyte-like cell line Raji, do not promote Th1 responses.

3. Similarly, is the down-modulation of Gata-3 and T-bet specific to NK EVs?

The effects of the NK-EV miRNAs miR-92a and miR-10b on CD4^+^T cells were analyzed, showing similar effects on GATA-3 and T-BET expression levels by western blot than incubation with NKEVs (Figure 3—figure supplement 1E). As indicated in the previous points, other EVs (HEK or Raji) or non-NK-EV miRNAs (hsa-miR-124, hsa-miR-3667 and hsa-miR-4158, new Figure 6—figure supplement 2), had no effect on T cell function.

4. EVs are produced by many cell types. Is it possible to estimate the proportion of NK-EVs in total EVs?

Estimating the proportion of a specific subset of EVs would be very interesting, but quite challenging technically. Different approaches, such as the use of specific markers for serum analyses (e.g. perforin, Granzyme B) may be addressed, however this would not allow to discriminate from other cytotoxic cell-derived EVs, and the amounts of these vesicles in serum from healthy donors is low for these kind of analyses.

Reviewer #2 (Recommendations for the authors):1) The authors isolated EVs with differential ultracentrifugation only which is prone to contamination of protein complexes, lipoparticles etc. Can the authors show EM data of their EVs.

As suggested by the reviewer, we have performed Electron microscopy from NK-EVs isolated from NK cell culture supernatants by differential ultracentrifugation, as described in Figure 1figure supplement 1A and methods. Electron microscopy confirmed the average size range of small EVs (between 50-200nm) and their typical cup-shape morphology and membranous structure by negative staining (Figure 1—figure supplement 1H). EM analyses showed isolated vesicles with little non-small EV material. The main text of the manuscript and the material and methods section have been modified accordingly.

2) Because the authors have not performed size-exclusion nor density gradient separation before applying their functional experiments with purified EVs they cannot rule out that other molecules in the activated NK secretome are responsible for the observed effects? Blocking EV uptake with dynasore for example could rule out a major contribution from other soluble factors.

First, we analyzed the effects of NK-derived EVs on isolated T lymphocytes pre-treated with Dynasore to block EV uptake. As included in the new figure (new Figure 3—figure supplement 2A,B and Figure 4—figure supplement 2A,B), after dynamin-dependent endocytosis blockade, the upregulation of IFN-γ and CD25 and driven by NK-EVs were abolished, indicating that the observed effects on T cell function are mediated by NK-derived EVs. It is worthwhile mentioning, that since dynasore has been shown to enhance the formation of mitochondrial antiviral signalling aggregates and promote endocytosis-independent NF-κB activation, this may interfere with lymphocyte activation (1).

To further rule out the involvement of soluble factors co-precipitating with the small EV pellet after 100,000 g ultracentifugation, we isolated small EVs from NK cell culture supernatants, using size-exclusion chromatography (SEC) (new Figure 3—figure supplement 2C-H and new Figure 4figure supplement 2C,D). Fractions 8-11, that expressed the exosomal markers CD81 and Tsg101 and low levels of protein aggregates (new Figure 3—figure supplement 2C,D), were pooled and ultracentrifuged before functional analysis. Despite a significant loss of NK-EV material during the SEC purification steps (new Figure 3—figure supplement 2E,F), confirmed by a reduction of the exosomal markers CD81 and Tsg101 to around 20% of the levels obtained by ultracentrifugation, a similar impact for SEC-isolated NK-derived small EVs on T cell function was observed, both in IFN-γ (Figure 3—figure supplement 2G,H) and more significantly in CD25 expression (Figure 4—figure supplement 2C,D).

Altogether, experiments using either Dynasore or SEC, confirmed that NK-EVs account for the observed effects, although we cannot completely rule out that the presence of soluble factors may also play a partial role. These experiments have been added to the manuscript as supplementary materials, as indicated, and the main text, discussion and material methods have been modified accordingly, as highlighted in the revised version of the manuscript.

3) The suppl data shows some western blotting of EVs from activated NK cells. Apart from their DE miRNA analysis between EVs from resting vs activated NK cells, it is possible that resting NK cells secrete much less EVs, it would behove the authors to demonstrate this with western and NTA or TPRS.

Resting NK cells were isolated form healthy donor´s buffy coats and, in parallel, activated NK cells were generated using our standard procedure, as described in the methods section. Small EVs were isolated from both resting and activated NK cells by ultracentrifugation, as described in the manuscript. It is important to highlight that resting CD3^-^CD56^+^ NK cells are a small proportion of peripheral-blood mononuclear cells, representing according to Angelo et al., 0.61 – 16.87% in healthy adults, with a mean of 6.47% (2). Thus, the number of resting NK cells that we could isolate from buffy coats were very limited (up to 5x10^6^ total cells) and therefore the amounts of small EVs that we could isolate were very low, even from activated NK cells, and unfortunately below the limit of detection of our western blotting techniques.

Nevertheless, using Nanoparticle-tracking analysis (NTA), we could detect small EVs derived from 5x10^6^ either resting or in vitro activated NK cells, although at very low levels, and confirmed that activated NK cells release significantly more small EVs than their resting cell counterparts (Figure 1—figure supplement 1I). Indeed, very few particles were detected in resting NK cell-derived small EVs. Moreover, dot blot analyses allowed to detect CD81 and Tsg101 exosomal markers in small EVs purified from activated NK cells, while EVs released by the same number of resting NK cells could not be detected (Figure 1—figure supplement 1J). Altogether our data confirm that resting NK cells release much less EVs than activated NK cells.

4) How do the authors exclude the presence of feeder cell EVs in their EV-preps?

Concerning the reviewer´s comment, we acknowledge that the presence of feeder cell-derived EVs in the vesicle preparation needs to be avoided. For this reason, our protocol rules out this possibility by isolating NK cells expanded in the presence of the cytokine cocktail and feeder cells, as described in methods, before small EV accumulation and isolation. After NK cell isolation, using the appropriate Miltenyi kit, following the manufacturer´s indications, the purity of isolated NK cells was checked by flow cytometry (CD3^-^ CD56^+^), before incubating the cells in EV-depleted medium for small EV accumulation and further isolation by ultracentrifugation. This has been further clarified in the main text of the revised manuscript.

5) Because the authors have not used control EVs from other cell types it remains somewhat unclear whether all EVs or specifically EVs from (activated) NK cells have the polarizing effects on the recipient cells. Perhaps the authors could use an irrelevant control cell type that does not express much less of their miRNAs. I do appreciate that the nanoparticle experiments strongly suggest the involvement of miRNAs (in vivo). The authors should use their nanogold particles with miRNAs as controls for some of the crucial the in vitro NK-EVs experiments. Do their nanoparticles have similar effects on CD4^+^ Th1 polarization and what candidate miRNAs have this effect? Ideally, the authors would examine the effects of NK-EVs in mice but I find that not required.

To address this key point several new experiments were performed.

First, small EVs from two distinct human cell lines (namely the HEK-293, human epithelial kidney cells and the Raji B lymphoblast cells) were isolated, following the differential ultracentrifugation protocol, as described in the methods section. Their effects in primary T cells isolated from human healthy donors showed no impact, neither in IFN-γ secretion (new Figure 3—figure supplement 3), nor in activation, measured by CD25 expression (Figure 4—figure supplement 2E,F), that even decreased upon Raji B cell EV-treatment under Th1-polarizing conditions.

Also, three microRNAs that are preferentially excluded from the NK-EV fraction were selected, namely hsa-miR-124, hsa-miR-3667 and hsa-miR-4158 and loaded onto gold-nanoparticles (new Figure 6—figure supplement 2), and their effects were evaluated in immunocompetent C57/BL6 mice after footpad injection. These experiments showed no effects of these nanoparticles, as observed for NK-EV enriched microRNAs, neither in activation, nor in IFN-γ secretion (new Figure 6H).

6) Maybe the authors can include a separate table of the miRNAs that change in the activated NKs cells?

Following the reviewer´s comment we have included a separate table where we show the relative abundance of EV-miRNAs in secreting activated NK cells and their secreted EVs from small RNA sequencing data, and the corresponding plots, including statistics (new Figure 1figure supplement 2B,C).

7) Log fold changes (alone) can be deceiving. Maybe the authors can show (relative) abundance data of their EV-enriched miRNAs?

Besides the new figure mentioned above, individual values can be found in Supplementary Table S1.

8) Template-independent terminal ribonucleotide transferases (TENTs) catalyze the addition of nucleotide monophosphates to the 3′-end of RNA molecules. While I appreciate their efforts in their pMT (I prefer isomiR) analysis, global analysis of all miRNAs are in my view not very informative. It is surprising that so many cytosine additions are found whereas adenylation and uridylation are known to be caused by uridylases and adenylases. Is there evidence of 3' tagging of mature mammalian miRNAs by cytidylation. Obviously, it would be interesting if this type of modification is somehow NK cell function related.

Following the reviewer´s suggestion we have modified in the text to refer to modified miRNAs as isomiRs. As the reviewer points out, uridylation and adenylation are the most typical 3′ end modifications across animal miRNAs (3-6).

Several studies evaluating PtMs have found guanidinylation and cytosilation to be barely represented. However, cytosine addition was found to be the second most abundant 3´modification in mouse primordial germ cells and gonadal somatic cells at various embryonic stages (7) and was also described in Arabidopsis (8).

Moreover, in a recent paper from our group, cytosylation was found to be a significant posttranscriptional modification in human T cells (9), suggesting that cytosine additions may be relevant for human lymphocyte function.

9) Despite a large body of independent evidence, the biological relevance of many Isomirs as found in sequencing data has recently been scrutinized (Hu et al., Nature Biotech 2021). Perhaps the authors can comment in their paper why their results are unlikely the result of artifacts during library preparation?

We acknowledge that ligation bias has been well documented in conventional small RNA-seq methods. It has thus been suggested that non-canonical “modification tails” may be an artifact due to the specific 3’ adaptor sequence and/or ligation chemistry used for library preparation.

In our case, the adaptor remains cannot contribute to the presence of non-templated "C" residues at the 3' end of miRNAs, as described in Rodríguez-Galan et al., (9). Briefly, in our experimental settings, adaptors used in the current experiments are standard Illumina TruSeq adaptors that start with the sequence "AGATCGGAAGAGCACACGTCT". Adaptor sequences were removed with cutadapt, using the following line of code, for each fastq file:

cutadapt -m 15 -M 35 -O 7-o trimmed.fq --untrimmed-output=untrimmed.fq --too-shortoutput=tooshort.fq -a AGATCGGAAGAGCACACGTCTGAACTCCAGTCAC input.fq

Regarding the possibility of ligation bias, we are aware that this may affect any library preparation strategy. While it is true that some bias might exist in the absolute quantification of miRNAs with a specific nucleotide composition, in our manuscript, we focus on comparing this composition across different types of samples (resting NK cells, activated NK cells and NK-derived EVs). Thus, this bias would be affecting every sample and our comparisons would still be valid.

10) Their analysis of the NTAs, though comprehensive, makes little sense to me. The authors state that they analyzed variations from position -4 to +4. If the average mature miRNA is 21nts long, they thus examined sequences of 17-25 nucleotides, yet in their methods, they stated that they 'filter' the reads from 20nts? Please clarify.

We agree with the reviewer that our analysis is based on non-templated modifications, being either substitutions within the canonical sequence (mainly positions -1, 0), or additions, from position +1 onwards. For the sake of clarity, we have modified the nomenclature in the main text and refer to each specific modification as either nucleotide “additions” or “substitutions”, as suggested by the reviewer.

The average size of mature miRNAs is 22 nt and more than 90% of those described in miRBAse21 have lengths equal to 20 nt and/or higher. For the current study, read pre-processing was performed with a pipeline that used cutadapt and FastQC: the first to remove Illumina adapters and to select trimmed reads with lengths equal or larger than 20, and the second to perform quality checks on the reads.

To identify PtMs, we used Chimira to align pre-processed reads against miRNA hairpin sequences from the human component of miRBase, with the aim of identifying nucleotide sequences diverging from the reference sequence (modifications).

To provide a common, unified framework for visualization and PtM summarization, Chimira adjusts the alignments on the 3'end, mapping everything to a length of 23 nt, using a zero-based coordinate system.

In that context, the initial position (5'end) is referred as "zero", and modification positions located before or after the 5'end are identified with negative and positive indices, respectively. The end position (3'end) is numbered as position "22", independently of the real length of miRNAs. As it happened on the 5'ends, modification positions located before or after the 3'end are identified with negative and positive indices, respectively.

Therefore, by focusing on positions -4 to +4, we were searching for sequence variations, relative to each reference sequence in miRBAse, in a 9 nt long region that was centered on the 3'end of miRNAs, independently of their length.

11) How is it possible that position 0 (the last nucleotide on the 3' end of the miRNA) has a nucleotide addition if anything it should be a substitution? The authors should show the percentage of fully mature (untagged) canonical miRNAs in their graphs. The first NTA should be position +1, etc.

Please refer to the previous point. We have modified the nomenclature as suggested by the reviewer as highlighted in the main manuscript text.

12) Figure 3B is interesting in that some miRNAs such miR122-5p is for 50% modified in resting cells yet not at all in activated cells, yet the (activated) NK cell EVs do contain modified miRNAs. What are the absolute or relative levels of these miRNAs in the NK cells? Can the authors provide an explanation?

We agree with the reviewer that it is interesting that certain miRNAs, namely miR-122-5p, but also miR-409-3p and miR-451a that follow a similar trend, are highly modified in resting NK cells, and in activated NK-derived EVs, while absent y activated NK cells. We believe, that a possible explanation for this observation is, that upon activation, modified miRNAs, e.g. miR-122-5p are more prone to be sorted into EVs. In this sense, it was described that uridylation may promote exosome loading, while adenylated miRNA isoforms were relatively enriched in B cells (10). Thus, a similar mechanism might be ocurrying in human activated NK cells for modified miRNAs, that could be tagged for EV packaging. This has been added to the Discussion section of the revised manuscript. Concerning the relative values of these miRNAs, for miR-122-5p, these levels are relatively low but very similar in both resting and activated NK cells, while high in NK-EVs (Relative Counts Per Million; REST: 0,57, ACT: 0,52 and NK-EVs: 313,94).

Reviewer #3 (Recommendations for the authors):In the report entitled "Natural killer (NK) cell-derived extracellular-vesicle shuttled microRNAs control T cell responses", Dosil et al., identified NK-extracellular-vesicle (EV)-associated microRNAs and their post-transcriptional modifications signature by small RNA next-generation sequencing. They also found that NK-EVs promoted Th1 polarization and activation of monocyte and moDCs. They suggested that the identified NK-EV-associated microRNAs partially recapitulated NK-EV effects in T cells in vivo.The study contains some interesting findings. However, the impacts of NK-EVs and NK-EV-associated microRNAs on Th1 differentiation are not impressive. In addition, their evidence is not sufficient to support their proposal that NK-EV-associated microRNAs promote Th1-like responses via T-bet de-repression by down-regulation of GATA3.1. The authors validated selected microRNAs expression by qPCR in Figure 1E. However, no microRNAs were significantly enriched in NK-EVs compared with activated NK cells. To support the author's proposal, statistical significance of their expression between activated NK cells and NK-EVs should be confirmed.

We thank the reviewer for highlighting this point. We have now included a separate table where we show the relative abundance of EV-miRNAs in secreting activated NK cells and their secreted EVs from small RNA sequencing data, and the corresponding plots, including statistics (new Figure 1—figure supplement 2B,C).

Also, we have re-analyzed the qPCR data and show Paired t-test analysis, comparing Resting vs Activated cells, Activated cells vs NK-EVs and Resting cells vs NK-EVs (Figure 1E,F). These analyses confirm the results from the sequencing data, showing differences between Activated NK cells and their released NK-EVs, that reach statistic significance in most cases. For hsa-miR-10b-5p and hsa-miR-99a-5p validations by qPCR, comparing activated cells and their released EVs, t-tests yields p-values slightly above 0.1 (0.1656 and 0.1302, respectively), due to the variability among donors. However, the enrichment of these miRNAs in NK-EVs compared to their parental cells is consistent within individuals, as shown in Figure 1—figure supplement 2 and Supplementary Table S1.

2. The authors demonstrated that NK-EV-associated microRNAs possessed unique post-transcriptional modifications and short motifs in Figure 2 and Figure 3. Since the conclusions from both figures are similar, it would be better to combine Figure 2 and Figure 3 together.

We have now merged these two figures as new Figure 2, and added the additional panels as figure supplements in the revised manuscript, and modified the text accordingly.

3. As mentioned above, the impact of NK-EVs on Th1 polarization was quite minor in Figure 4. Furthermore, the authors failed to induce IFN-γ production from CD4^+^ T cells by administration of selected microRNAs in Figure 7F.NK-EVs contain several effector proteins including IFN-γ (Choi et al., Molecules, 2020;25(21)). So, they should exclude the possibility of contamination of IFN-γ derived from NK-EVs in Figure 4.

To address this issue, and since ultracentrifugation of cell culture supernatants does not allow to fully rule out the involvement of other molecules, such as IFN-*γ* in the observed effects, we have performed additional experiments, using different approaches.

First, we analyzed the effects of NK-derived EVs on isolated T lymphocytes pre-treated with Dynasore to block EV uptake. As included in the new figure (new Figure 3—figure supplement 2A,B and Figure 4—figure supplement 2A,B), after dynamin-dependent endocytosis blockade, the upregulation of IFN-γ and CD25 and driven by NK-EVs were abolished, indicating that the observed effects on T cell function are mediated by NK-derived EVs. It is worthwhile mentioning, that since dynasore has been shown to enhance the formation of mitochondrial antiviral signalling aggregates and promote endocytosis-independent NF-κB activation, this may interfere with lymphocyte activation (1).

To further rule out the involvement of soluble factors co-precipitating with the small EV pellet after 100,000 g ultracentifugation, we isolated small EVs from NK cell culture supernatants, using size-exclusion chromatography (SEC) (new Figure 3—figure supplement 2C-H and new Figure 4figure supplement 2C,D). Fractions 8-11, that expressed the exosomal markers CD81 and Tsg101 and low levels of protein aggregates (new Figure 3—figure supplement 2C,D), were pooled and ultracentrifuged before functional analysis. Despite a significant loss of NK-EV material during the SEC purification steps (new Figure 3—figure supplement 2E,F), confirmed by a reduction of the exosomal markers CD81 and Tsg101 to around 20% of the levels obtained by ultracentrifugation, a similar impact for SEC-isolated NK-derived small EVs on T cell function was observed, both in IFN-γ (Figure 3—figure supplement 2G,H) and more significantly in CD25 expression (Figure 4—figure supplement 2C,D).

Altogether, experiments using either Dynasore or SEC, confirmed that NK-EVs account for the observed effects, although we cannot completely rule out that the presence of soluble factors may also play a partial role. These experiments have been added to the manuscript as supplementary materials, as indicated, and the main text, discussion and material methods have been modified accordingly, as highlighted in the revised version of the manuscript.

4. The authors proposed that NK-EV-associated microRNAs promote Th1-like responses via T-bet de-repression by downregulating Gata3 mRNA. To ensure their proposal, double-staining of T-bet and GATA3 by flow cytometry should be conducted in Figure 4 and Figure 7.

To address this point raised by the reviewer two strategies have been carried out. First, we have analyzed the effects of the addition of a mixture of hsa-miR-92a-3p and hsa-miR-10b-5p AuNPs to CD4^+^ T cells, that putatively target GATA-3 (Supplementary Table 2). Three days after incubation, we found that, as observed for mRNA transcript levels (Figure 3E), GATA-3 protein levels were reduced while T-bet expression increased, compared to CD4^+^ T cells incubated with Control-AuNPs by western blot (Figure 3—figure supplement 1F).

Also, a similar trend was observed by flow cytometry analyses after NK-EV incubation (Figure 3figure supplement 1E), although GATA-3 levels were low. However, since it is well documented that miRNAs target several mRNAs, very likely other targets may contribute to the observed effects.

5. In Figure 8, the authors represent that NK-EV-associated microRNAs promote Th1-like responses. However, it seems to be an overestimation because they did not show the downregulation of GATA3 by NK-EV-associated microRNAs and T-bet de-repression as well as IFN-γ induction from CD4^+^ T cells by treatment of NK-EV-associated microRNAs. This fails to support the story of this manuscript.

Please refer to the previous point.

References

M. Ailenberg, C. Di Ciano-Oliveira, K. Szaszi, Q. Dan, M. Rozycki, A. Kapus, O. D. Rotstein, Dynasore enhances the formation of mitochondrial antiviral signalling aggregates and endocytosis-independent NF-kappaB activation. British journal of pharmacology 172, 3748-3763 (2015); published online EpubAug (10.1111/bph.13162).

L. S. Angelo, P. P. Banerjee, L. Monaco-Shawver, J. B. Rosen, G. Makedonas, L. R. Forbes, E. M. Mace, J. S. Orange, Practical NK cell phenotyping and variability in healthy adults. Immunol Res 62, 341-356 (2015); published online EpubJul (10.1007/s12026-015-8664y).

A.M. Burroughs, Y. Ando, M. J. de Hoon, Y. Tomaru, T. Nishibu, R. Ukekawa, T. Funakoshi, T. Kurokawa, H. Suzuki, Y. Hayashizaki, C. O. Daub, A comprehensive survey of 3' animal miRNA modification events and a possible role for 3' adenylation in modulating miRNA targeting effectiveness. Genome Res 20, 1398-1410 (2010); published online EpubOct (10.1101/gr.106054.110).

H. R. Chiang, L. W. Schoenfeld, J. G. Ruby, V. C. Auyeung, N. Spies, D. Baek, W. K. Johnston, C. Russ, S. Luo, J. E. Babiarz, R. Blelloch, G. P. Schroth, C. Nusbaum, D. P. Bartel, Mammalian microRNAs: experimental evaluation of novel and previously annotated genes. Genes and development 24, 992-1009 (2010); published online EpubMay 15 (10.1101/gad.1884710).

P. Landgraf, M. Rusu, R. Sheridan, A. Sewer, N. Iovino, A. Aravin, S. Pfeffer, A. Rice, A. O. Kamphorst, M. Landthaler, C. Lin, N. D. Socci, L. Hermida, V. Fulci, S. Chiaretti, R. Foa, J. Schliwka, U. Fuchs, A. Novosel, R. U. Muller, B. Schermer, U. Bissels, J. Inman, Q. Phan, M. Chien, D. B. Weir, R. Choksi, G. De Vita, D. Frezzetti, H. I. Trompeter, V. Hornung, G. Teng, G. Hartmann, M. Palkovits, R. Di Lauro, P. Wernet, G. Macino, C. E. Rogler, J. W. Nagle, J. Ju, F. N. Papavasiliou, T. Benzing, P. Lichter, W. Tam, M. J. Brownstein, A. Bosio, A. Borkhardt, J. J. Russo, C. Sander, M. Zavolan, T. Tuschl, A mammalian microRNA expression atlas based on small RNA library sequencing. Cell 129, 1401-1414 (2007); published online EpubJun 29 (10.1016/j.cell.2007.04.040).

H. Muller, M. J. Marzi, F. Nicassio, IsomiRage: From Functional Classification to Differential Expression of miRNA Isoforms. Frontiers in bioengineering and biotechnology 2, 38 (2014) (10.3389/fbioe.2014.00038).

D. M. Kruger, S. Neubacher, T. N. Grossmann, Protein-RNA interactions: structural characteristics and hotspot amino acids. Rna 24, 1457-1465 (2018); published online EpubNov (10.1261/rna.066464.118).

M. T. Chou, B. W. Han, C. P. Hsiao, P. D. Zamore, Z. Weng, J. H. Hung, Tailor: a computational framework for detecting non-templated tailing of small silencing RNAs. Nucleic acids research 43, e109 (2015); published online EpubSep 30 (10.1093/nar/gkv537).

Rodriguez-Galan, S. G. Dosil, M. J. Gomez, I. Fernandez-Delgado, L. FernandezMessina, F. Sanchez-Cabo, F. Sanchez-Madrid, MiRNA post-transcriptional modification dynamics in T cell activation. iScience 24, 102530 (2021); published online EpubJun 25 (10.1016/j.isci.2021.102530).

D. Koppers-Lalic, M. Hackenberg, I. V. Bijnsdorp, M. A. J. van Eijndhoven, P. Sadek, D. Sie, N. Zini, J. M. Middeldorp, B. Ylstra, R. X. de Menezes, T. Wurdinger, G. A. Meijer, D. M. Pegtel, Nontemplated nucleotide additions distinguish the small RNA composition in cells from exosomes. Cell reports 8, 1649-1658 (2014); published online EpubSep 25 (10.1016/j.celrep.2014.08.027).

R. Garcia-Martin, G. Wang, B. B. Brandao, T. M. Zanotto, S. Shah, S. Kumar Patel, B. Schilling, C. R. Kahn, MicroRNA sequence codes for small extracellular vesicle release and cellular retention. Nature 601, 446-451 (2022); published online EpubJan (10.1038/s41586-021-04234-3).